# C3DM: Constrained-Context Conditional Diffusion Models for Imitation Learning

**Vaibhav Saxena**                                      *vsaxena33@gatech.edu*
*School of Interactive Computing*
*Georgia Institute of Technology*

**Yotto Koga**                                          *yotto.koga@autodesk.com*
*Robotics Lab*
*Autodesk Research*

**Danfei Xu**                                           *danfei@gatech.edu*
*School of Interactive Computing*
*Georgia Institute of Technology*

**Reviewed on OpenReview:** *https://openreview.net/forum?id=jcleXdnRA1*

## Abstract

Behavior Cloning (BC) methods are effective at learning complex manipulation tasks. However, they are prone to *spurious correlation*—expressive models may focus on distractors that are irrelevant to action prediction—and are thus fragile in real-world deployment. Prior methods have addressed this challenge by exploring different model architectures and action representations. However, none were able to balance between sample efficiency and robustness against distractors for solving manipulation tasks with a complex action space. We present **C**onstrained-**C**ontext **C**onditional **D**iffusion **M**odel (C3DM), a diffusion model policy for solving 6-DoF robotic manipulation tasks with robustness to distractions that can learn deployable robot policies from as little as five demonstrations. A key component of C3DM is a *fixation* step that helps the action denoiser to focus on task-relevant regions around a predicted *fixation point* while ignoring distractors in the context. We empirically show that C3DM is robust to out-of-distribution distractors, and consistently achieves high success rates on a wide array of tasks, ranging from table-top manipulation to industrial kitting that require varying levels of precision and robustness to distractors.[1]

## 1 Introduction

Behavior cloning (BC) is a simple yet effective method for learning from offline demonstrations when such data is available. However, with limited training data and high-capacity neural network models, a BC policy trained to map high-dimensional input such as images to actions often degenerates to focusing on spurious features in the environment instead of the task-relevant ones (De Haan et al., 2019). This leads to poor generalization and fragile execution in real-world applications such as kitting and assembly, where acting in scenes with distractors is inevitable. This challenge is especially prominent for continuous-control problems, where instead of committing to one target, the policy often collapses to the "average" of a mix of correct and incorrect predictions due to distractions. Our research aims to develop a class of sample-efficient BC methods that are robust to distractions in the environment.

Prior work attempts to address this challenge through different model architectures and action representations. A prominent line of research (Zeng et al., 2021; Shridhar et al., 2022a;b) uses fully convolutional networks (FCNs) with residual connections that 1) learn mappings from visual input

---

[1]Project website: `https://sites.google.com/view/c3dm-imitation-learning`.

to a *spatially-quantized action space*, and 2) attend to *local features* in addition to a global view of the input. In these models each possible action gets assigned with a probability mass, which avoids the action averaging effect making the model less prone to bad precision due to distractions.

While effective for 2D tasks such as pushing piles and transporting objects, this strategy prohibits tasks with complex 6-DoF action spaces as the quantization space grows exponentially with action dimensions. For modeling actions in a continuous space, generative policy models (Florence et al., 2021; Weng et al., 2023; Chi et al., 2023) have shown that flexible probabilistic models such as Diffusion Models (Ho et al., 2020; Chi et al., 2023) allow for modeling complex actions. However, the underlying high-capacity neural network models are still prone to learning spurious input-output correlations when training data is limited.

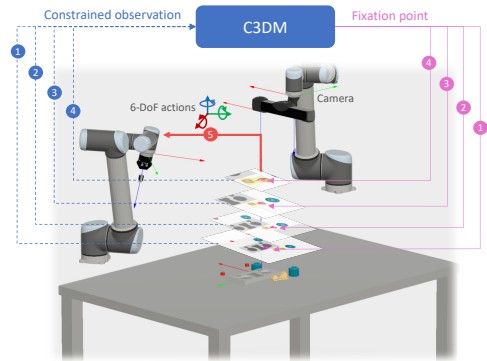

In this work, we argue that these generative policy models can be made robust to distractions if we imbue them with the ability to attend to *inferred* local features, like an FCN, and share parameters across the processing of such *constrained contexts*. Hence, we build a conditional generative model that *controls what it sees* in the input context by inferring action-relevant parts of the scene using a "fixation point" parameterization, and train it to iteratively refine its action prediction as it also refines its

Figure 1: Illustrating 6-DoF action prediction for table-top manipulation using our diffusion model (C3DM), which implicitly learns to fixate on relevant parts of the input and iteratively refines its prediction using action-relevant details about the observation.

input context. Our method, **C**onstrained-**C**ontext **C**onditional **D**iffusion **M**odel (C3DM), is a diffusion model policy that solves 6-DoF robotic manipulation tasks with high sample efficiency and robustness to distractions. Our model learns a distribution over actions given a global input context as either an RGB image or a depth map. Similar to (Chi et al., 2023), our inference distribution over actions is fixed to a Gaussian noising process, but in addition to that we also infer *constrained observations* that are fixated around the target action. The learned generative distribution utilizes an iteratively refining denoising process on both the observation and action variables. Our novel denoising process implements "fixation" by *zooming* into a part of the input image, or *masking* out pixels, around iteratively-predicted "fixation points" at each denoising iteration. This procedure, that we call the *f*ixation-while-Denoising Diffusion Process (*f*DDP) facilitates invariance to distractions during action denoising. We illustrate this process in Figure 1 with the detailed procedure for implementing the proposed method in Figure 2. Our key contributions are:

1. We propose a theoretically-grounded fixation-while-Denoising Diffusion Process (*f*DDP) framework that enables a Diffusion Policy (Chi et al., 2023) to progressively "zoom into" action-relevant input throughout the iterative denoising process.

2. We show that our method, C3DM, outperforms existing generative policy methods on a wide array of simulated tasks, such as sweeping, sorting, kitting and assembly, deploy our model on a real robot to perform sorting and insertion after training on just 20 demonstrations, and demonstrate robust sim-to-real transfer on a kitting and cup hanging task from 5 demonstrations.

## 2 Related Works

**Visual Imitation Learning.** Imitation learning (IL) has been proven effective for robotic manipulation tasks (Schaal, 1999; Billard et al., 2008; Englert & Toussaint, 2018). Recent works (Zhang et al., 2018; Finn et al., 2017; Mandlekar et al., 2020; 2021; Brohan et al., 2022) use deep neural networks to map directly from image observations to action, and demonstrated visuomotor learning for complex and diverse manipulation tasks. However, these policies tend to generalize poorly to new situations due to spurious connections between pixels and actions (Wang et al., 2021; De Haan et al., 2019) and requires extensive training data to become robust. To address the challenge, recent works (Abolghasemi et al., 2018; Wang et al., 2021; Zhu et al.,

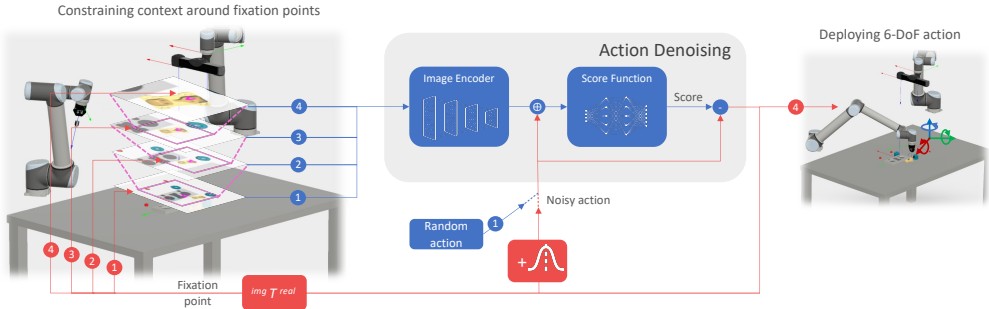

Figure 2: **C**onstrained-**C**ontext **C**onditional **D**iffusion **M**odel (C3DM) for visuomotor policy learning. Here we illustrate the iterative action-refinement procedure (4 timesteps) using our fixation-while-denoising diffusion process ($f$DDP), wherein we constrain our input context around a "fixation point" predicted by the model ($\wedge$) at each refinement step. Subsequently, we refine the predicted action by fixating only on the useful part of the context, hence removing distractions and making use of higher levels of detail in the input.

2022) have incorporated visual attention Mnih et al. (2014) as inductive biases in the policy. For example, VIOLA (Zhu et al., 2022) uses object bounding priors to select salient regions as policy input. Our constrained context formulation can be similarly viewed as a visual attention mechanism that iteratively refines with the diffusion process. Another line of work exploits the spatial invariance of fully convolutional models and discretized action space to improve learning sample efficiency (Zeng et al., 2021; Shridhar et al., 2022a;b). Notably, PerAct (Shridhar et al., 2022b) show strong performance on 6DoF manipulation tasks. However, to implement discretized 3D action space, PerAct requires a large voxel-wise transformer model that is expensive to train and evaluate, and it also requires 3D point cloud input data. Our method instead adopts an implicit model that can model arbitrary action space and only requires 2D image input.

**Diffusion Policy Models.** Diffusion models have shown remarkable performance in modeling complex data distributions such as high-resolution images (Song et al., 2020; Ho et al., 2020). More recently, diffusion models have been applied to decision making (Chi et al., 2023; Ajay et al., 2022; Zhong et al., 2023; Mishra et al., 2023) and show promising results in learning complex human actions (Chi et al., 2023; Pearce et al., 2023; Reuss et al., 2023) and conditionally generating new behaviors (Ajay et al., 2022; Zhong et al., 2023). Closely related to our work is DiffusionPolicy (Chi et al., 2023) that uses a diffusion model to learn visuomotor policies. However, despite the demonstrated capability in modeling multimodal actions, DiffusionPolicy still learns an end-to-end mapping between the input and the score function and is thus prone to spurious correlation, as we will demonstrate empirically. Our method introduces a framework that enables the denoising diffusion process to fixate on relevant parts of the input and iteratively refine its predictions.

**Implicit Policy Models.** Closely related to diffusion policy models are implicit-model policies (Florence et al., 2021; Jarrett et al., 2020), which represent distributions over actions using Energy-Based Models (EBMs) (LeCun et al., 2006). Finding optimal actions with an energy function-based policies can be done through sampling or gradient-based procedures such as Stochastic gradient Langevin dynamics. Alternatively, actions prediction can be implicitly represented as a distance field to the optimal action (Weng et al., 2023). Similar to diffusion policies, implicit models can represent complex action distributions but are also prone to spurious correlations in imitation, as we will demonstrate empirically.

## 3  Background

**Denoising Diffusion Probabilistic Models (DDPMs)** are a class of generative models that introduce a hierarchy of latent variables and are trained to maximize the variational lower-bound on the log-likelihood of observations (ELBO). To compute the ELBO, this class of models fixes the inference distribution on the introduced latent variables to a *diffusion process*, that can be aggregated to directly infer each latent variable from the observation. Say $\mathbf{x}_0$ represents the random variable that we want to model, we introduce $T$ latent variables $\{\mathbf{x}_t\}_{t=1}^{T}$ each of which can be inferred from the observation $\mathbf{x}_0$ using

$$q(\mathbf{x}_t|\mathbf{x}_0) := \sqrt{\overline{\alpha}(t)} \cdot \mathbf{x}_0 + \sqrt{1 - \overline{\alpha}(t)} \cdot \epsilon_t, \tag{1}$$

where $\epsilon_t \sim \mathcal{N}(\mathbf{0}, I)$ is the standard normal Gaussian noise, and $\overline{\alpha}(t)$ is a fixed "noise schedule" that determines the variance of the noising process, and satisfies $\overline{\alpha}(t) < 1 \ \forall t$ and $\overline{\alpha}(t) \rightarrow 0$ as $t \rightarrow T$. We call the index $t$ "time," and in principle the cardinality of the set $\{\mathbf{x}_t\}_{t=1}^T$ should tend to infinity. The choice of $q(\mathbf{x}_t|\mathbf{x}_0)$ can play a significant role in the learning process of the generative distribution, and we will see in this paper how it affects the performance of our action prediction model during evaluation. Additionally, the denoising process can be obtained using the Bayes' rule as $q(\mathbf{x}_{t-1}|\mathbf{x}_t, \mathbf{x}_0) = q(\mathbf{x}_t|\mathbf{x}_{t-1}, \mathbf{x}_0)\frac{q(\mathbf{x}_{t-1}|\mathbf{x}_0)}{q(\mathbf{x}_t|\mathbf{x}_0)}$.

Each latent variable $\mathbf{x}_1, \ldots, \mathbf{x}_T$ can be interpreted as a noisy variant of $\mathbf{x}_0$, and the generative distribution over $\mathbf{x}_0$ is modeled using conditional distributions $p(\mathbf{x}_{t-1}|\mathbf{x}_t)$ as $p(\mathbf{x}_0) = p(\mathbf{x}_T)\prod_{t=1}^T p(\mathbf{x}_{t-1}|\mathbf{x}_t)$, where $p(\mathbf{x}_T)$ is fixed to be a standard Gaussian prior to comply with the noising process in (1). The process of learning $p(\mathbf{x}_0)$ comes down to minimizing divergences $\text{KL}(q(\mathbf{x}_{t-1}|\mathbf{x}_t, \mathbf{x}_0)||p(\mathbf{x}_{t-1}|\mathbf{x}_t))$ to learn $p(\mathbf{x}_{t-1}|\mathbf{x}_t)$. As shown by Ho et al. (2020), we can learn $p$ by training an auxiliary "score function" $f_\theta$, parameterized using a neural network, that is (generally) trained to predict the standard normal noise $\epsilon_t$ that was used to infer $\mathbf{x}_t$ ($\epsilon$-prediction), that is, $\theta = \text{argmin}_\theta MSE(f_\theta(\mathbf{x}_t, t), \epsilon_t)$. Other design choices include training $f_\theta$ to predict the denoised sample $\mathbf{x}_0$ directly (x-prediction), or a combination of $\mathbf{x}$ and $\epsilon$ (v-prediction) (Salimans & Ho, 2022).

To generate samples of $\mathbf{x}_0$, we begin with samples of $\mathbf{x}_T$, which for the chosen noising process in (1) should result in a sample from the standard normal distribution. Then, we "denoise" $\mathbf{x}_T$ using the $\epsilon_T$ noise predicted by the learned score function $f_\theta(\mathbf{x}_T, T)$ to obtain the next latent $\mathbf{x}_{T-1} \sim p(\mathbf{x}_{T-1}|\mathbf{x}_T)$. This is one step of the "iterative refinement" process. We continue to refine the sample for $T$ steps after which we return the last denoised sample, which is a sample from the distribution $p(\mathbf{x}_0)$.

**Problem Setup (Imitation Learning).** In this work, we focus on modeling a conditional generative distribution over 12-dimensional action variables ($\mathbf{x}_0 \in \mathbb{R}^{12}$) conditioned on image observations $\mathbf{O}$, i.e. $p(\mathbf{x}_0|\mathbf{O})$. We obtain observations from a downward-facing camera over a table-top, and output two 6-DoF gripper poses as actions for a robot, which comprise 3D position and Euler rotation angles around *z-x-y* axes (in that order) in the camera frame. The output actions then parameterize primitive pick-and-place motions that can be deployed on a robot using any off-the-shelf motion planner to generate a trajectory leading to the 6-DoF gripper poses output from the generative model.

## 4 Method

We present a **C**onstrained-**C**ontext **C**onditional **D**iffusion **M**odel (C3DM) that predicts robot actions conditioned on image observations. Our novel *f*ixation-while-Denoising Diffusion Process (*f*DDP) infers a "fixation point" in the input at each action-refinement step, which it iteratively uses to 1) ignore distractions by constraining its input context around that point, and 2) improve its precision by querying for higher levels of detail in the image input around the predicted fixation point. *f*DDP exploits the observation-action coupling which allows it to determine fixation points in the context and constrain its observation throughout the action refinement process.

### 4.1 Conditional Diffusion for Action Prediction

Diffusion models have shown great success in image and video generation (Ho et al., 2020; 2022; Song et al., 2020). Recently their conditional counterpart (Diffusion Policy (Chi et al., 2023)) has been shown to learn stable image-conditioned robot policies with little hyperparameter tuning. A major contributing factor for the success of these models is the iterative refinement procedure that *reuses* model capacity to *iteratively denoise* latent samples in the action space into useful ones that match the data distribution. More precisely, these models learn a denoiser function $\epsilon_\theta$ that learns to denoise a random action $\mathbf{a}_T$ into a target action $\mathbf{a}_0$ given a *static* observation $\mathbf{O}$ in an iterative fashion. This iterative refinement process can be depicted as, $\mathbf{a}_T \xrightarrow{\epsilon_\theta(\ \cdot\ ;\ \mathbf{O})} \mathbf{a}_{T-1} \xrightarrow{\epsilon_\theta(\ \cdot\ ;\ \mathbf{O})} \cdots \xrightarrow{\epsilon_\theta(\ \cdot\ ;\ \mathbf{O})} \mathbf{a}_0$. However, given limited offline demonstrations, these models are not immune from learning *spurious correlations* between image input ($\mathbf{O}$) and action output ($\mathbf{a}$), which generally leads to imprecise and fragile execution on robots. We argue that while it is hard to control what correlations a model learns from data, it is possible to control what the model *sees* and use it to our advantage.

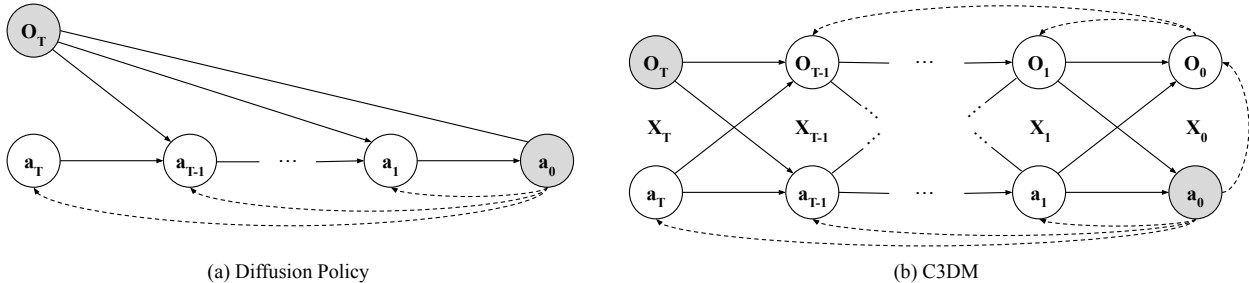

(a) Diffusion Policy          (b) C3DM

Figure 3: (a) Action inference and generation in **Diffusion Policy,** compared with (b) inference and generation of observation-action tuples in **C3DM.** Filled circles represent observed variables. In (b), solid lines represent the generative distributions, $p_\theta(\mathbf{O}_{t-1}|\mathbf{X}_t)$ for generating latent (fixated) observations and $p_\theta(\mathbf{a}_{t-1}|\mathbf{X}_t)$ for the next latent (noisy) actions. Solid and dashed lines together show the inference (de-noising) distributions, $q(\mathbf{O}_{t-1}|\ \mathbf{O}_t, \mathbf{X}_0)$ and $q(\mathbf{a}_{t-1}|\ \mathbf{X}_t, \mathbf{a}_0)$ for inferring latent observations and actions respectively.

In this paper, we build a diffusion model that learns to iteratively constrain its input context around inferred action-relevant parts of the scene – in a process we call "fixation." As we will discuss later, this process can also be viewed as a form of data augmentation that further boosts the learning sample efficiency. In summary, our model 1) learns to infer action-relevant parts of the scene as "fixation-points" while performing action denoising, and 2) appends the iterative action refinement with observation refinement that we implement as *zooming* or *masking* around the inferred fixation-point, that helps improve robustness and sample efficiency in action prediction.

## 4.2 Constrained-Context Conditional Diffusion Model (C3DM)

We propose a $f$ixation-while-Denoising Diffusion Process ($f$DDP) that utilizes a joint iterative refinement of intermediate observations and actions (deemed *fixated* and *noisy* respectively). In the following subsections, we formulate the different components that make up the $f$DDP, essentially the generative and inference distributions, and talk about our specific implementations of the same.

**Generative distribution.** We build a theoretical framework for generating target action $\mathbf{a}_0$, given a single global observation $\mathbf{O}_T$ of the scene, by introducing both latent observations and actions for timesteps 1 to $T-1$. We define the joint distribution for such generation as,

$$p_\theta(\mathbf{a}_0|\mathbf{O}_T) = \int p_\theta(\mathbf{a}_{0:T}, \mathbf{O}_{0:T-1}|\mathbf{O}_T) d\mathbf{a}_{1:T} d\mathbf{O}_{0:T-1}, \tag{2}$$

where $\{\mathbf{a}_t\}_{t=1}^T$ are noisy actions and $\{\mathbf{O}_t\}_{t=0}^{T-1}$ are fixated observations. We define a generated tuple of fixated observation and action as $\mathbf{X}_t := (\mathbf{a}_t, \mathbf{O}_t)$. We factorize the joint as a reverse de-noising process given by,

$$p_\theta(\mathbf{a}_{0:T}, \mathbf{O}_{0:T-1}|\mathbf{O}_T) = p_\theta(\mathbf{a}_T|\mathbf{O}_T) \prod_{t=1}^{T} p_\theta(\mathbf{X}_{t-1}|\mathbf{X}_t). \tag{3}$$

We further break down the generation $p_\theta(\mathbf{X}_{t-1}|\mathbf{X}_t)$ into,

$$\begin{aligned} p_\theta(\mathbf{X}_{t-1}|\mathbf{X}_t) &= p_\theta(\mathbf{O}_{t-1}|\mathbf{X}_t) p_\theta(\mathbf{a}_{t-1}|\mathbf{O}_{t-1}, \mathbf{X}_t) \\ &= \underbrace{p_\theta(\mathbf{O}_{t-1}|\mathbf{X}_t)}_{\text{fixated obs}} \underbrace{p_\theta(\mathbf{a}_{t-1}|\mathbf{X}_t)}_{\text{next noisy action}}, \end{aligned} \tag{4}$$

where $p_\theta(\mathbf{a}_{t-1}|\mathbf{X}_t)$ generates the "next noisy action" and $p_\theta(\mathbf{O}_{t-1}|\mathbf{X}_t)$ the next "fixated observation." Given a global observation of the scene, we can use these generative distributions to iteratively generate denoised actions and fixated observations where the latter act as input to the next action denoising. This framework hence enables action denoising with a different observation input at each timestep, which in this work we leverage for action generation with robustness to distractors. We illustrate these generative distributions as solid lines in Figure 3.

**Inference distribution.** We formalize the inference distribution as a reverse de-noising process $q(\mathbf{X}_{t-1}|\mathbf{X}_t, \mathbf{X}_0)$, which we break down as,

$$q(\mathbf{X}_{t-1}|\mathbf{X}_t, \mathbf{X}_0) = q(\mathbf{a}_{t-1}, \mathbf{O}_{t-1}|\ \mathbf{X}_t, \mathbf{X}_0)$$
$$= \underbrace{q(\mathbf{O}_{t-1}|\ \mathbf{X}_t, \mathbf{X}_0)}_{\text{fixated obs}} \underbrace{q(\mathbf{a}_{t-1}|\ \mathbf{X}_t, \mathbf{a}_0)}_{\text{next noisy action}}. \tag{5}$$

We compute the fixated observation $q(\mathbf{O}_{t-1}|\ \mathbf{O}_t, \mathbf{X}_0)$ using either a zooming or masking process centered around a fixation point that we obtain from the given action $\mathbf{a}_0$ by exploiting the observation-action alignment in the setup. The latent actions $q(\mathbf{a}_{t-1}|\ \mathbf{O}_{t-1}, \mathbf{a}_0)$ are sampled using a fixed noising process, which we discuss in detail in Section 4.3.1. The role of these inference distributions is to help train the generative distributions we obtained in (4) by providing a way to generate fixated observations and noisy actions that are privy of the ground truth action, which our training procedure then attempts to match with the parameterized generative distributions in (4) (see Figures 3 and 12).

Note that we directly define the denoising process in $q$, whereas DDPM first defines the noising process as a Gaussian and uses Bayes' rule to obtain the denoising process in closed form. We do this because our noising process to get fixated observations would be implemented as an *unmask* or *zoom-out* process whose inverse cannot be written in closed form, leading us to resort to directly defining the denoising distributions in $q$.

**Training objective.** The likelihood $p_\theta(\mathbf{a}_0|\mathbf{O}_T)$ can be maximized by minimizing the KL divergence between the fixed inference distributions $q(\mathbf{X}_{t-1}|\mathbf{X}_t, \mathbf{X}_0)$ and learned $p_\theta(\mathbf{X}_{t-1}|\mathbf{X}_t) \forall t$. Please see proof in Appendix A.2.

### 4.3 Implementing C3DM

### 4.3.1 Action Noising (and the choice of noising process)

During training, for each target action $\mathbf{a}^{(i)}$, where $i$ is the sample index in the training dataset, we sample $K$ "noisy" actions $\{\tilde{\mathbf{a}}_k^{(i)}\}_{k=1}^K$ by adding noise vectors using a fixed noising process, conditioned on timesteps $\{t_k\}_{k=1}^K$, $t_k \sim \text{Unif}(0, T)$ (we use $T = 1$ in our setup). Note that this implements the noisy action inference $q(\mathbf{a}_{t-1}|\ \mathbf{X}_t, \mathbf{a}_0)$ in (5).

Fixation, as we will see later, not only changes the observation, but also the coordinate frame of the action at each refinement iteration. This introduces an additional challenge to training the denoising model which has to infer the noise level for different denoising steps in addition to generalizing to different observation fixation levels. To address this challenge, we introduce a simple yet powerful modification to the standard noising process - we remove the drifting of the target action and rather diffuse the action only when obtaining noisy (latent) actions. This lets us control the noisy action to always stay within bounds that are observable for the model in the fixated observation, making action denoising significantly easier to learn for different fixation levels. We formalize both these processes below.

**Noising process with drift.** The standard noising process in DDPM can be written as,

$$\tilde{\mathbf{a}}_k^{(i)} = \sqrt{\overline{\alpha}(t_k)} \cdot \mathbf{a}^{(i)} + \sqrt{1 - \overline{\alpha}(t_k)} \cdot \epsilon_k^{(i)}, \tag{6}$$

where $\epsilon_k^{(i)} \sim \mathcal{N}(\mathbf{0}, I)$ with $\mathcal{N}$ being a normal distribution and $I$ the identity matrix. $\overline{\alpha}(t_k)$ is a tunable noise schedule that satisfies the conditions specified in Section 3.

**Noising process without drift (ours).** Our noising process diffuses but does not drift the noisy actions, and can be written as,

$$\tilde{\mathbf{a}}_k^{(i)} = \mathbf{a}^{(i)} + \sqrt{1 - \overline{\alpha}(t_k)} \cdot \epsilon_k^{(i)}. \tag{7}$$

As we show in Section 5.1.2, we find the noising process in (7) (no drift) to perform better empirically than the one with drift, and hence is the default setting for all our main results. We also point out that using (7) would result in $\mathbf{a}_k$ being sampled from a uniform random distribution when $t_k = T$ rather than a standard normal distribution, as is the case when using (6). See detailed proof in Appendix A.3.2.

### 4.3.2 Fixation and Context Constraining

We imbue our diffusion model with the ability to implicitly identify action-relevant regions and ignore distractions in the input as it iteratively refines actions. We do this by determining a "fixation point" in

the observation at each step of the action refinement procedure. We exploit the alignment between the observation and action space and set the fixation point to the target actions during training. This echos with many prior works (e.g., Zeng et al. (2021); Shridhar et al. (2022b)) that exploit this observation-action alignment for improving sample efficiency.

**Fixation.** The fixation point for each observation image is a function of the target action, given by $\mathbf{p}^{(i)} := {}^{\text{img}}T^{\text{real}}\text{pos}(\mathbf{a}^{(i)})$, where ${}^{\text{img}}T^{\text{real}}$ is a matrix that transforms points in the camera frame to image frame, and $\text{pos}(\mathbf{a}^{(i)})$ extracts the 2-D position of actuation on the x-y plane from the input 6-DoF action. The constrained context can then be formulated as $\mathbf{O}_k^{(i)} := C(\mathbf{O}^{(i)}; \mathbf{p}^{(i)}, t_k)$, where $C$ either masks parts of the observation far away from $\mathbf{p}^{(i)}$ (C3DM-Mask) or zooms into the context fixated at $\mathbf{p}^{(i)}$ (C3DM). Note that context constraining implements the fixated observation inference $q(\mathbf{O}_{t-1}|\mathbf{X}_t, \mathbf{X}_0)$ formulated in (5). We implement $p_\theta(\mathbf{O}_{t-1}|\mathbf{X}_t)$ (in (4)) during testing by using the intermediate denoised action for the fixation point.

**Constraining context by masking (C3DM-Mask).** For masking an image of size $H \times W$, we first create a window of size $\frac{H * t_k}{T} \times \frac{W * t_k}{T}$ centered at the fixation point $\mathbf{p}^{(i)}$, add random jitter to the window location while still keeping $\mathbf{p}^{(i)}$ inside the window to minimize the train-test gap where the next fixation point might not necessarily be at the center of the constrained context. Then, we mask all pixels outside this window with the background value. We illustrate this in Figure 4(top).

**Constraining context by zooming (C3DM).** Similar to fixation by masking, we first create a window around the fixation point $\mathbf{p}^{(i)}$ with random jitter to the window location. Then, we use a cached high-resolution image of the same scene to obtain a zoomed-in image in the constrained window. Our new context is still of the same input size ($H \times W$) but now has higher levels of detail. We note that while we do assume access to high-res images, our model does not utilize the entire image but rather only task-relevant parts for action inference. Not using the entire high-res image keeps the model activation size small and inference time as low as possible for this method (given the additional computational

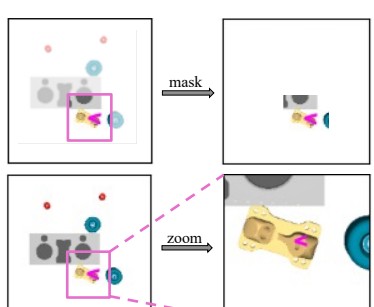

Figure 4: Illustrating masking (top) and zooming (bottom) for constraining context around predicted fixation point ($\wedge$).

overload introduced due to observation encoding at each denoising timestep). While we utilize cached high-res images during training, one can use this model to physically move the camera closer to the fixation point to obtain and process higher levels of detail in the input during inference. We illustrate this process in Figure 4(bottom).

After zooming in to obtain the constrained context, we re-normalize the noisy action $\tilde{\mathbf{a}}_k^{(i)}$ to the new camera frame for input to the denoiser network during training. This ensures that the noisy action and constrained observation input to the denoiser are consistent. The denoiser output is still expected in the unconstrained context to keep the scale of the model output constant.

***Baked-in data augmentation.*** C3DM trains an action denoiser given a variety of constrained contexts $\mathbf{O}_k^{(i)}$ over and above the training set of images. Additionally, with our zooming-in approach, we also transform the ground-truth action into new camera frames. This essentially increases the coverage of both the observation and action spaces seen during training and deems our method very sample efficient.

### 4.3.3 Action Denoising

We learn robot policies in a table-top setup that has a single downward-facing camera to obtain top-down observations of the table. For each observation $\mathbf{O}^{(i)}$ in the dataset we obtain latent encodings $\mathbf{o}^{(i)} := \text{enc}_\phi(\mathbf{O}^{(i)})$, where $\text{enc}_\phi$ is parameterized using a ResNet-18 (He et al., 2015). We build a model $\epsilon_\theta$ parameterized by a fully-connected MLP that is supervised to predict the standard normal noise vector ($\epsilon\text{-prediction}$) given the embeddings of the constrained input observation $\mathbf{o}_k^{(i)}$ and noisy action $\tilde{\mathbf{a}}_k^{(i)}$. That is, we train our model using the MSE loss,

$$\mathcal{L}(D) := \frac{1}{N}\frac{1}{K}\sum_{i=1}^{N}\sum_{k=1}^{K} ||\epsilon_\theta(\mathbf{o}_k^{(i)}, \tilde{\mathbf{a}}_k^{(i)}) - \epsilon_k^{(i)}||^2. \tag{8}$$

We summarize the training procedure in Algorithm 1.

### 4.3.4 *f*ixation-while-Denoising Diffusion Process for Action Generation (*f*DDP)

The context constrainer $C$, observation encoder $\text{enc}_\phi$, and the action denoiser $\epsilon_\theta$ in conjunction with the reversible noising process in (7), together facilitate the fixation-while-Denoising Diffusion Process (*f*DDP) that we use for generating action given any observation input. We summarize our iterative observation and action refinement process that implements the generative distributions $p_\theta(\mathbf{O}_{t-1}|\mathbf{X}_t)$ and $p_\theta(\mathbf{a}_{t-1}|\mathbf{X}_t)$ respectively (in (4)) in Algorithm 2. Precisely, we implement context-constraining in $p_\theta(\mathbf{O}_{t-1}|\mathbf{X}_t)$ around intermediate denoised actions for action-relevant fixation, obtain a sequence of latent actions and constrained contexts $(\mathbf{O}_T, \mathbf{a}_T) \to (\mathbf{O}_{T-1}, \mathbf{a}_{T-1}) \to \cdots \to (\mathbf{O}_0, \mathbf{a}_0)$, and return the last denoised action $\mathbf{a}_0$ as predicted action. We provide an illustration in Figure 2, and an example showing fixation points and fixated observations in *f*DDP in Figure 6.

---

**Algorithm 1** C3DM - Training *f*DDP components

**Require:** $D = \{\mathbf{O}^{(i)}, \mathbf{a}^{(i)}\}_{i=1}^N, K, \text{max\_iters}, ^{\text{img}}T^{\text{cam}},$
$\quad \phi, \text{enc}_\phi, \theta, \epsilon_\theta, \text{T}$
$\quad$ **for all** n\_iter $\in \{1, \ldots, \text{max\_iters}\}$ **do**
$\qquad L \leftarrow 0$
$\qquad$ **for all** $k \in \{1, \ldots, K\}$ **do**
$\qquad\quad$ **for all** $i \in \{1, \ldots, N\}$ **do**
$\qquad\qquad t_k \sim \text{Unif}(0, \text{T})$ $\qquad\qquad$ ▷ sampled timestep
$\qquad\qquad \epsilon_k^{(i)} \sim \mathcal{N}(\mathbf{0}, I)$ $\qquad\qquad$ ▷ sampled noise
$\qquad\qquad \tilde{\mathbf{a}}_k^{(i)} \leftarrow \mathbf{a}^{(i)} + \sqrt{1 - \overline{\alpha}(t_k)} \cdot \epsilon_k^{(i)}$ ▷ noisy action
$\qquad\qquad \mathbf{p}^{(i)} \leftarrow {}^{\text{img}}T^{\text{real}}\, \text{pos}(\mathbf{a}^{(i)})$ $\quad$ ▷ fixation point
$\qquad\qquad \mathbf{O}_k^{(i)} \leftarrow C(\mathbf{O}^{(i)}; \mathbf{p}^{(i)}, t_k)$ ▷ constrained context
$\qquad\qquad \mathbf{o}_k^{(i)} \leftarrow \text{enc}_\phi(\mathbf{O}_k^{(i)})$ ▷ encoding constrained obs
$\qquad\qquad L \leftarrow L + ||\epsilon_\theta(\mathbf{o}_k^{(i)}, \tilde{\mathbf{a}}_k^{(i)}) - \epsilon_k^{(i)}||^2$
$\qquad\quad$ **end for**
$\qquad$ **end for**
$\qquad \phi \leftarrow \phi - \frac{1}{NK}\nabla_\phi L$ $\qquad$ ▷ update encoder params
$\qquad \theta \leftarrow \theta - \frac{1}{NK}\nabla_\theta L$ $\qquad$ ▷ update denoiser params
$\quad$ **end for**

---

**Algorithm 2** C3DM - Iterative refinement with *f*DDP for action prediction during testing

**Require:** $\mathbf{O}_T, \text{act\_bounds}, ^{\text{img}}T^{\text{cam}}, \phi, \text{enc}_\phi, \theta, \epsilon_\theta, \text{T}$
$\quad \mathbf{a_T} \sim \text{Unif}(\text{act\_bounds})$
$\quad$ **for all** $t \in \{\text{T}, \ldots, 1\}$ **do**
$\qquad \mathbf{o}_t \leftarrow \text{enc}_\phi(\mathbf{O}_t)$ $\qquad\qquad$ ▷ encoding constrained obs
$\qquad \mathbf{a}_t^0 \leftarrow \mathbf{a}_t - \sqrt{1 - \overline{\alpha}(t)} \cdot \epsilon_\theta(\mathbf{o}_t, \mathbf{a}_t)$ $\quad$ ▷ action denoising
$\qquad \epsilon \sim \mathcal{N}(\mathbf{0}, I)$ $\qquad\qquad$ ▷ sampled noise
$\qquad \mathbf{p}_t \leftarrow {}^{\text{img}}T^{\text{real}}\, \text{pos}(\mathbf{a}_t^0)$ $\qquad$ ▷ fixation point
$\qquad \mathbf{a}_{t-1} \leftarrow \mathbf{a}_t^0 + \sqrt{1 - \overline{\alpha}(t-1)} \cdot \epsilon$ $\quad$ ▷ noising using (7)
$\qquad \mathbf{O}_{t-1} \leftarrow C(\mathbf{O}_t, \mathbf{p}_t)$ $\qquad$ ▷ fixated obs for next iter
$\quad$ **end for**
$\quad$ **return** $\mathbf{a}_0$

---

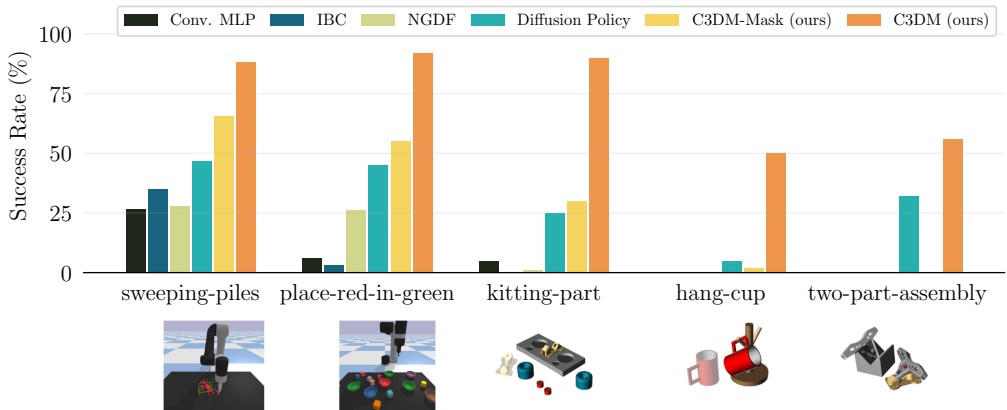

Figure 5: (Top) Success rates for manipulation tasks in simulation (average across 100 rollouts, peak performance in 500 epochs of training. (Bottom) Illustration of the simulated evaluation tasks.

## 5   Experiments

We evaluated C3DM extensively on 5 tasks in simulation and demonstrated our model on various tasks with real robot arms. For the simulated tasks, we used Implicit Behavioral Cloning (IBC, Florence et al. (2021)), Neural Grasp Distance Fields (NGDF, Weng et al. (2023)), Diffusion Policy (Chi et al., 2023), and an explicit behavior cloning model (Conv. MLP) with the same backbone as our method for the baselines. Additionally, we ablated against the masking version of C3DM, called C3DM-Mask. Each task assumed access to 1000 demonstrations from an oracle demonstrator. For the real robot evaluation, we used Diffusion Policy as the baseline. We used 20 human demonstrations in one set of real robot experiments and just 5 simulated demonstrations for the sim-to-real experiments. We used RGB inputs for all simulation and real robot experiments except for when performing sim-to-real transfer where we used depth maps to reduce the domain gap between sim and real. Hence, we also highlight here that our method can be used to solve tasks irrespective of the input modality.

### 5.1   Simulation Experiments

#### 5.1.1   Tasks

We evaluated our method on 5 different tasks in simulation (see Table 5 and Appendix B for a detailed description of each task). Since IBC experimented predominantly on tasks with "push" primitives, we included the *sweeping-piles* task for a fair comparison. The remaining tasks, *place-red-in-green, hang-cup, kitting-part,* and *two-part-assembly* are based on "pick/grasp" and "place" primitives. *Place-red-in-green* and *sweeping-piles* were made available by Zeng et al. (2020) as part of the Ravens simulation benchmark, and we built the remaining tasks on the robotics research platform provided by Koga et al. (2022). An oracle demonstrator provided 1000 demonstrations for all tasks, where each demonstration is a unique reset of the task containing a single top-down observation of the scene and the pick-place action (no intermediate waypoint observations). The success rates for the various methods are shown in Figure 5.

#### 5.1.2   Main results

**C3DM is sample efficient and learns policies from minimal data.**   We tested the sample efficiency of C3DM against the Diffusion Policy baseline and found that our method was able

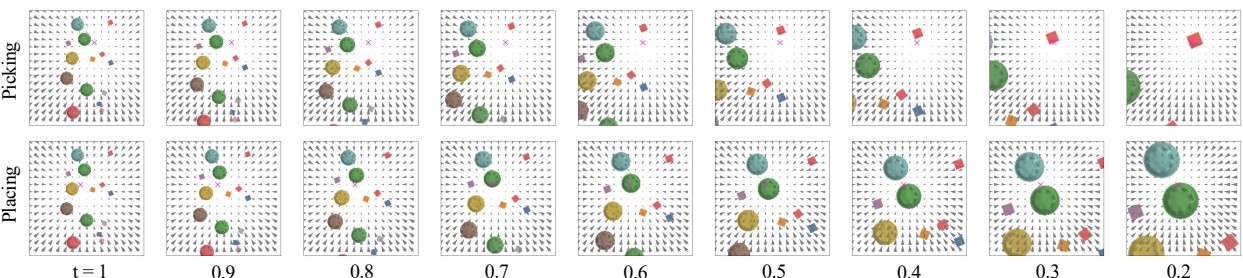

Figure 6: **Fixation-while-Denoising Diffusion Process (ƒDDP).** This figure shows score fields when inferring the pick (top) and place (bottom) actions for the *place-red-in-green* task using C3DM. The score field (output by $\epsilon_\theta$, depicted here as a 2D projection on the input image) at each timestep induces a fixation point denoted by × (that can be obtained by probing the field at a random point, translating towards the predicted direction, and transforming into the image space). The input image at each timestep (other than $t = 1$) is obtained by zooming around the fixation point (×) predicted at the previous timestep. For both picking and placing, we observe that the model is distracted at first with the entire table-top in view. As it zooms in (with decreasing timestep), our model *fixates* closer and closer to the objects of interest (red block for picking and green bowl for placing) showcasing a less distracted prediction owing to the removal of distractors and higher spatial resolution utilized during the denoising process.

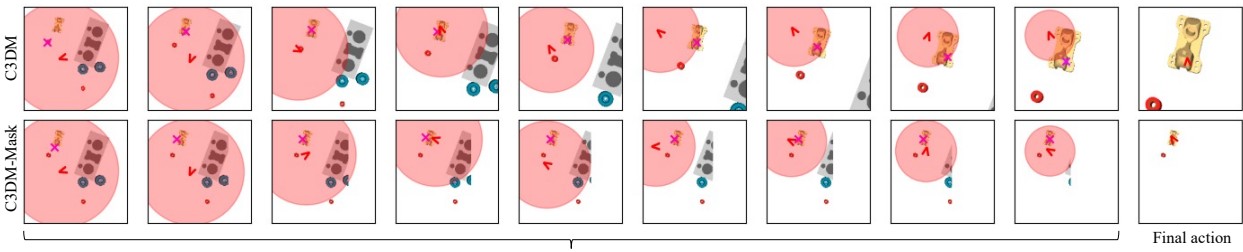

Figure 7: **C3DM vs C3DM-Mask.** Context refinement by fixation in conjunction with action denoising using $f$DDP on the *kitting-part* task (picking) (C3DM (top) and C3DM-Mask (bottom)). The red region depicts the standard deviation of the predicted latent action, $\wedge$ represents the latent action (orientated with grasp yaw), and $\times$ is the fixation point. The observation at each refinement iteration is fixated around the fixation point ($\times$) predicted in the previous timestep, using a zooming mechanism for C3DM and masking for C3DM-Mask. As refinement progresses, we observe how the model fixates on the target part (yellow) as it refines its action for grasping.

to achieve good performance using just 30 demos in simulation, reaching ~90% success on the *place-red-in-green* sorting task while the baseline achieved very little success, as shown in Figure 8. We attribute this to the high observation and action space coverage of our method that clearly helps with sample efficiency.

**C3DM can ignore distractions in table-top manipulation.** Both C3DM-Mask and C3DM showed good performance on *place-red-in-green* due to their ability to ignore distractor objects leading to precise locations for picking. On the contrary, baseline methods struggled to fixate on the target object that needed to be picked. In *kitting-part*, while baseline methods were able to predict actions approximately around the target action, they were not precise enough to succeed given the low tolerance of this task. C3DM ignored unnecessary objects on the table, as shown in Figure 7, leading to precise pick and place predictions. We observed failures when the distractor objects were too close to the object and fixation in C3DM was unable to minimize their effect.

**C3DM is invariant to unseen distractions.** To test the generalization ability of C3DM against unseen distractors, we added unseen objects usually found on table-tops such as a pen, remote, mouse, etc. (shown in Figure 13), all made available in the SAPIEN dataset (Xiang et al., 2020; Mo et al., 2019; Chang et al., 2015). We report success rates comparing C3DM and Diffusion Policy in Table 2. We found a <10% drop in success rate of C3DM when replacing seen distractors with unseen ones suggesting very good generalization to ignoring unseen distractors compared to Diffusion Policy.

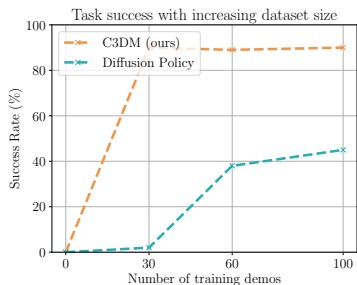

Figure 8: Sample efficiency of C3DM (peak success rates in 500 epochs of training).

Table 1: Model performance, with and without drift in the diffusion process, after training for 200 epochs on *place-red-in-green* task.

| Method | C3DM | C3DM-Mask |
|---|---|---|
| **Drift** (6) | 60% | 40% |
| **No drift** (7) | **79%** | **47%** |

**Action refinement with a fixated gaze can help predict 6-DoF gripper poses with high precision.** C3DM is the only method that could succeed substantially on the *hang-cup* and *two-part-assembly* tasks due to its ability to precisely predict the full 6-DoF action. C3DM beat all other baselines and the C3DM-Mask ablation showing the importance of action refinement with a fixated gaze. We illustrate how our learned score field creates this fixation in Figure 6. The cases where our model did fail were in which the model fixated on spurious locations early in the refinement process leading to the correct pick location being outside the region of view, as well as when small errors in the action prediction led to irrecoverable scenarios.

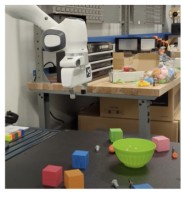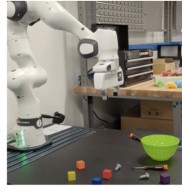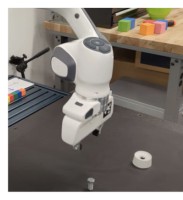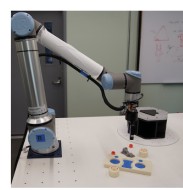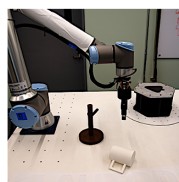

Figure 10: Real-world experiment setup and tasks, from left to right: place-block-in-bowl (big) place-block-in-bowl (small), screw-insert, kitting-part and hang-cup. First three were setup on a Franka Emika Panda robot and trained on human demos, the last two on a UR10 robot and deployed sim-to-real.

**Iterative refinement with more levels of input detail can solve visual challenges in tasks.** The *sweeping-piles* task poses the challenge of inferring the location of small particles and the target zone from an image, for which C3DM outperforms all considered baselines. We note that IBC's performance on this task is lower than that reported in their work for planar sweeping, because our version of this task is more visually complex with the target zone marked by a thin square as opposed to being marked by a color-filled region. The *kitting-part* task also presents a complex-looking part to infer the pose of in order to predict a stable grasp. C3DM can iteratively refine its pose by obtaining higher levels of detail in the input leading to a substantially low grasp error of 0.72 cm and 11.28° and

Table 2: Task success rates with unseen distractors (showing average over 3 random seeds and 1 standard deviation, across 20 rollouts, after 500 train epochs) on *place-red-in-green* task. Models saw blocks as distractors during training; we replaced them with daily-life objects (see Figures 13 and 14) to test generalization.

| Method | 50% unseen | 100% unseen |
|---|---|---|
| **Diffusion Policy** | $41.7\% \pm 5\%$ | $33.3\% \pm 10\%$ |
| **C3DM (ours)** | **$85\% \pm 4\%$** | **$90\% \pm 4\%$** |

eventually a higher success rate compared to all baseline methods, including the C3DM-Mask ablation, which can only ignore distractions in the input.

**Additional observations.** Table 1 shows comparison between two diffusion processes for training our model, one that drifts the latent action to the origin (Equation (6)), and the other a pure diffusion (Equation (7)). We observed the latter to perform better in practice and is the default choice for all our main results. We also show the success rates when varying the number of refinement steps during action refinement in Figure 9, and as expected we observe a rising trend in success as number of refinement steps increase.

### 5.2 Real-robot Experiments

We demonstrated our model's ability to ignore distractions and precise action prediction using two real robots. On a Franka Emika robot, we deployed policies trained on human-driven demonstration, and used an Intel RealSense D435 camera mounted to view the workspace top-down. On a UR10 robot,

Figure 9: Success rate on the *place-red-in-green* task with increasing number of refinement timesteps.

we deployed sim-to-real policies, with a MechEye Pro M depth camera that produced depth maps comparable to those in the simulator. Testing in both setups proved that our method can work reliably when trained on real images as well as be robust to the sim-to-real gap, making it extremely practical for real-world deployment.

#### 5.2.1 Model Performance with Real-world Demonstrations

For the setup that trains on real-world demonstrations, we collected 20 human demonstrations using a space-mouse teleoperator for the *place-block-in-bowl* and *screw-insert* tasks which required picking target objects and placing them into a goal location. In the sim-to-real setup, we collected up to 100 demonstrations for the *kitting-part* and *hang-cup* tasks (as described in Appendix B) in simulation using a scripted policy.

Table 3: Success rates for tasks on a real Franka Emika robot.

| Method | place-block-in-bowl | | screw-insert | |
|---|---|---|---|---|
| | big blocks | small blocks | pick | pick+place |
| **Diffusion Policy** | 45% | 0% | 5% | 0% |
| **C3DM-Mask (ours)** | **60%** | 40% | 0% | 0% |
| **C3DM (ours)** | 55% | **65%** | **80%** | **60%** |

Each demo consisted of an initial image observation and the locations for pick and place actions in the robot's coordinate frame. We trained the baseline Diffusion Policy model, C3DM-Mask, and C3DM for 200 epochs. During testing, we evaluated all models using a linear schedule of 10 timesteps. All success rates averaged over 20 trials are summarized in Table 3 and Table 4.

**C3DM can ignore distractions in real images.** We performed experiments on two variants of the *place-block-in-bowl* task, one with big blocks with edge size 4.5 cm and other with small blocks of edge size 2.5 cm. Example image observations are shown in Figure 10. We observed that C3DM was able to effectively identify and ignore distractions in the input leading to precise actions for both variants. While C3DM-Mask was able to ignore distractions well, it lacked precision when evaluated with smaller blocks.

**Iterative context constraining leads to precise actions and high sample efficiency.** We performed experiments on a *screw-insert* task where the robot picks up a hex-head screw of edge size 1 cm, base diameter 1 cm, and places it in a receptacle with a hole of diameter 1.5 cm. Successfully completing this task required high precision as a pick prediction that is slightly off from the center of the screw would result in the screw falling on the table instead of being clamped by the parallel-jaw gripper of the robot. We observed that C3DM was significantly more successful in completing this task. While other methods failed due to both imprecise action prediction as well as the lack of

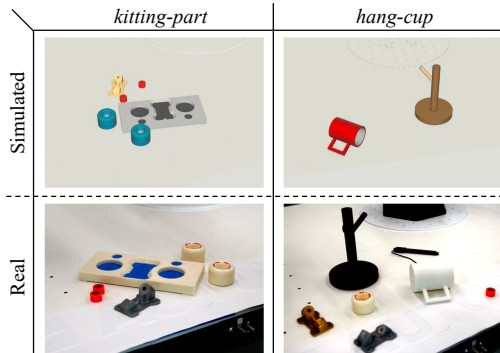

Figure 11: **Sim-to-real experiment.** Both real setups mimic the tasks we designed in sim. For *hang-cup* we evaluated C3DM with unseen distractors to further prove the robustness of our method even with the sim-to-real domain gap.

generalization capability given a small number of demonstrations, C3DM was able to generalize to all test locations as well as be precise within the 1 cm tolerance needed for this task.

### 5.2.2 Model Performance in Sim-to-Real

We conducted sim-to-real experiments using depth maps converted to height map observations of the *kitting-part* and *hang-cup* tasks. We used depth maps since the MechEye Pro M depth camera can produce depth map observations comparable to the simulator.

**C3DM can predict precise actions even with the sim-to-real domain gap.** Table 4 shows the success rate results over 20 trials of C3DM and the baseline Diffusion Policy for the 2 tasks. Both tasks required high precision for success with roughly 1-2 mm of positional and 1-2 degree rotational tolerance. The precision constraint for the cup hanging task is for the cup pick sub-task, where the clearance between the open fingers and the cup for grasping was roughly 1 mm. We found 5 demonstrations was sufficient to train C3DM in simulation

Table 4: Success rates for sim-to-real tasks on a UR10 robot.

| Method | #demos | kitting-part | hang-cup |
|---|---|---|---|
| **Diffusion Policy** | 5 | 0% | 0% |
| | 50 | 0% | 65% |
| | 100 | 0% | 60% |
| **C3DM (ours)** | 5 | **100%** | **100%** |

and deploy on the real robot setup with 100% success with random arrangements of the items. With 100 demonstrations, we could not get the deployed model to work for the *kitting-part* task with Diffusion Policy. The model worked well in simulation, so we attribute the slight noise difference in the real depth maps of the

distractor objects for the failure. It took 50 demonstrations to get the *hang-cup* task to work reasonably well with Diffusion Policy. Unexpectedly, the simulated success results for these tasks using RGB input (Figure 5) are lower than these results. We speculate that height maps are a better observation signal for certain tasks and hope to investigate this further in future work.

**C3DM can handle unseen distractors in real.** We evaluated C3DM and Diffusion Policy on the *hang-cup* task where the training data did not contain any distractors while the evaluation rollouts did. We found that C3DM showed reasonable performance with unseen distractor objects in the real deployment as shown in Figure 11 however Diffusion Policy showed no task success.

## 6 Limitations and Conclusion

We presented a **C**onstrained-**C**ontext **C**onditional **D**iffusion **M**odel (C3DM) for visuomotor imitation learning that learns to fixate on action-relevant parts of the input context while denoising actions. We demonstrated that the fixation-based context constraining allows our diffusion model to remove distractors from the input, achieving high success rate in a wide range of tasks requiring varying levels of precision and robustness against distraction. We also showed that our method can be deployed on a real robot for pick-and-place tasks either directly (sim-to-real) or by training on a handful of human demonstrations.

A limitation of our method is the task specific labelling of the fixation point in the demonstrations and selection of the final cropping dimension. For our set of examples, we set the fixation point at the finger location for the task (action). However, in other tasks, the region of interest may be elsewhere, for example, inserting a portion of a shape that is far from the grasp. The final cropping dimension was chosen to give an "ideal" view of our tasks (e.g., for picking tasks, the final cropping had the target object roughly fitting the view). We also want to point out that our method trades off precise action prediction and robustness against distractors for computational time. Since Diffusion Policy conditions on a single observation input, it can cache visual encodings for reuse across all denoising timesteps, however our method needs to encode observations at each refinement timestep. Another limitation, was the overhead camera, whose view of the target parts was easily occluded by the robot arms. This prevented seamless closed-loop error recovery. We plan to solve this by incorporating multiple camera views, such as an eye-in-hand camera configuration. We believe that future work with goal-conditioning policies and training on action trajectories (rather than sub-task goals) will render our model applicable for longer horizon manipulation.

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

# A Method Details

## A.1 Pictographic representation of the denoising pathway in C3DM

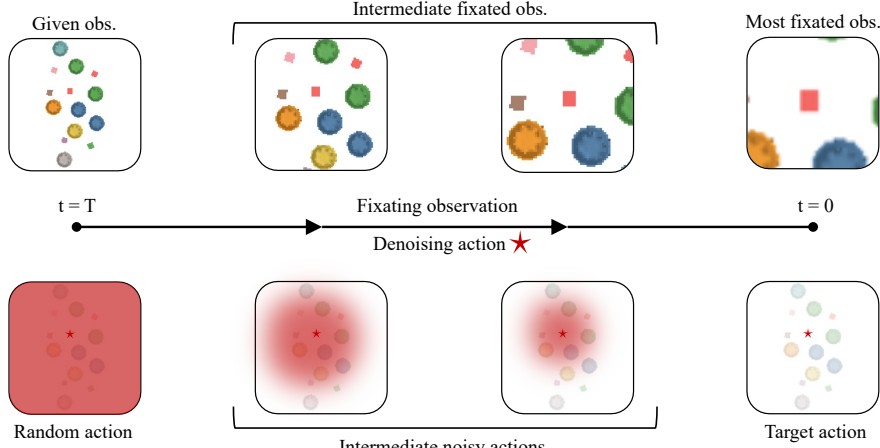

Figure 12: Illustrating action denoising without drift (bottom) and corresponding fixation in the observation (top). Fixation in the observation is implemented as a *zooming* process that directs camera observations towards a zoomed-in view around the target action.

## A.2 Proof of method

Given a global view of the scene $\mathbf{O}_T$, we train a generative model $p_\theta$ (formulated in (4)) that aims to generate target action $\mathbf{a}_0$ given the global observation. We do this by minimizing the cross-entropy loss which is given by,

$$L_{\text{CE}} = -\mathbb{E}_{q(\mathbf{a}_0|\mathbf{O}_T)} \log p_\theta(\mathbf{a}_0|\mathbf{O}_T)$$

$$= -\mathbb{E}_{q(\mathbf{a}_0|\mathbf{O}_T)} \log \left( \int p_\theta(\mathbf{a}_{0:T}, \mathbf{O}_{0:T-1}|\mathbf{O}_T) d\mathbf{a}_{1:T} d\mathbf{O}_{0:T-1} \right)$$

$$= -\mathbb{E}_{q(\mathbf{a}_0|\mathbf{O}_T)} \log \left( \int q(\mathbf{a}_{1:T}, \mathbf{O}_{0:T-1}|\mathbf{a}_0, \mathbf{O}_T) \frac{p_\theta(\mathbf{a}_{0:T}, \mathbf{O}_{0:T-1}|\mathbf{O}_T)}{q(\mathbf{a}_{1:T}, \mathbf{O}_{0:T-1}|\mathbf{a}_0, \mathbf{O}_T)} d\mathbf{a}_{1:T} d\mathbf{O}_{0:T-1} \right)$$

$$= -\mathbb{E}_{q(\mathbf{a}_0|\mathbf{O}_T)} \log \left( \mathbb{E}_{q(\mathbf{a}_{1:T}, \mathbf{O}_{0:T-1}|\mathbf{a}_0, \mathbf{O}_T)} \frac{p_\theta(\mathbf{a}_{0:T}, \mathbf{O}_{0:T-1}|\mathbf{O}_T)}{q(\mathbf{a}_{1:T}, \mathbf{O}_{0:T-1}|\mathbf{a}_0, \mathbf{O}_T)} \right)$$

$$\leq -\mathbb{E}_{q(\mathbf{a}_{0:T}, \mathbf{O}_{0:T-1}|\mathbf{O}_T)} \log \left( \frac{p_\theta(\mathbf{a}_{0:T}, \mathbf{O}_{0:T-1}|\mathbf{O}_T)}{q(\mathbf{a}_{1:T}, \mathbf{O}_{0:T-1}|\mathbf{a}_0, \mathbf{O}_T)} \right)$$

$$= -\mathbb{E}_{q(\mathbf{X}_{0:T-1}, \mathbf{a}_T|\mathbf{O}_T)} \log \left( \frac{p_\theta(\mathbf{X}_{0:T-1}, \mathbf{a}_T|\mathbf{O}_T)}{q(\mathbf{X}_{1:T-1}, \mathbf{a}_T, \mathbf{O}_0|\mathbf{a}_0, \mathbf{O}_T)} \right)$$

$$= \mathbb{E}_{q(\mathbf{X}_{0:T-1}, \mathbf{a}_T|\mathbf{O}_T)} \left[ -\log p_\theta(\mathbf{a}_T|\mathbf{O}_T) + \left( \log q(\mathbf{O}_0|\mathbf{a}_0, \mathbf{O}_T) + \sum_{t-1}^{T-1} \log \frac{q(\mathbf{X}_t|\mathbf{X}_{t-1}, \mathbf{O}_T)}{p_\theta(\mathbf{X}_{t-1}|\mathbf{X}_t)} + \log \frac{q(\mathbf{a}_T|\mathbf{X}_{T-1}, \mathbf{O}_T)}{p_\theta(\mathbf{X}_{T-1}|\mathbf{X}_T)} \right) \right]$$

$$= \mathbb{E}_{q(\mathbf{X}_{0:T-1}, \mathbf{a}_T|\mathbf{O}_T)} \left[ -\log p_\theta(\mathbf{a}_T|\mathbf{O}_T) + \left( \log q(\mathbf{O}_0|\mathbf{a}_0, \mathbf{O}_T) + \left[ \sum_{t-2}^{T-1} \log \frac{q(\mathbf{X}_t|\mathbf{X}_{t-1}, \mathbf{O}_T)}{p_\theta(\mathbf{X}_{t-1}|\mathbf{X}_t)} + \log \frac{q(\mathbf{a}_T|\mathbf{X}_{T-1}, \mathbf{O}_T)}{p_\theta(\mathbf{X}_{T-1}|\mathbf{X}_T)} \right] \right. \right.$$
$$\left. \left. + \log \frac{q(\mathbf{X}_1|\mathbf{X}_0, \mathbf{O}_T)}{p_\theta(\mathbf{X}_0|\mathbf{X}_1)} \right) \right]$$

$$= \mathbb{E}_{q(\mathbf{X}_{0:T-1}, \mathbf{a}_T|\mathbf{O}_T)} \left[ -\log p_\theta(\mathbf{a}_T|\mathbf{O}_T) + \left( \log q(\mathbf{O}_0|\mathbf{a}_0, \mathbf{O}_T) + \left[ \sum_{t-2}^{T-1} \log \frac{q(\mathbf{X}_{t-1}|\mathbf{X}_t, \mathbf{X}_0, \mathbf{O}_T)}{p_\theta(\mathbf{X}_{t-1}|\mathbf{X}_t)} \cdot \frac{q(\mathbf{X}_t|\mathbf{X}_0, \mathbf{O}_T)}{q(\mathbf{X}_{t-1}|\mathbf{X}_0, \mathbf{O}_T)} \right. \right. \right.$$
$$\left. \left. \left. + \log \frac{q(\mathbf{X}_{T-1}|\mathbf{a}_T, \mathbf{X}_0, \mathbf{O}_T)}{p_\theta(\mathbf{X}_{T-1}|\mathbf{X}_T)} \cdot \frac{q(\mathbf{a}_T|\mathbf{X}_0, \mathbf{O}_T)}{q(\mathbf{X}_{T-1}|\mathbf{X}_0, \mathbf{O}_T)} \right] + \log \frac{q(\mathbf{X}_1|\mathbf{X}_0, \mathbf{O}_T)}{p_\theta(\mathbf{X}_0|\mathbf{X}_1)} \right) \right]$$

$$
\begin{aligned}
&= \mathbb{E}_{q(\mathbf{X}_{0:T-1},\mathbf{a}_T|\mathbf{O}_T)}\Bigg[ -\log p_\theta(\mathbf{a}_T|\mathbf{O}_T) + \Bigg( \log q(\mathbf{O}_0|\mathbf{a}_0,\mathbf{O}_T) + \Bigg[ \sum_{t=2}^{T-1} \log \frac{q(\mathbf{X}_{t-1}|\mathbf{X}_t,\mathbf{X}_0,\mathbf{O}_T)}{p_\theta(\mathbf{X}_{t-1}|\mathbf{X}_t)} \\
&\qquad + \log \frac{q(\mathbf{X}_{T-1}|\mathbf{X}_0,\mathbf{O}_T)}{q(\mathbf{X}_1|\mathbf{X}_0,\mathbf{O}_T)} + \log \frac{q(\mathbf{X}_{T-1}|\mathbf{a}_T,\mathbf{X}_0,\mathbf{O}_T)}{p_\theta(\mathbf{X}_{T-1}|\mathbf{X}_T)} + \log \frac{q(\mathbf{a}_T|\mathbf{X}_0,\mathbf{O}_T)}{q(\mathbf{X}_{T-1}|\mathbf{X}_0,\mathbf{O}_T)} \Bigg] + \log \frac{q(\mathbf{X}_1|\mathbf{X}_0,\mathbf{O}_T)}{p_\theta(\mathbf{X}_0|\mathbf{X}_1)} \Bigg) \Bigg] \\
&= \mathbb{E}_{q(\mathbf{X}_{0:T-1},\mathbf{a}_T|\mathbf{O}_T)}\Bigg[ -\log p_\theta(\mathbf{a}_T|\mathbf{O}_T) + \Bigg( \log q(\mathbf{O}_0|\mathbf{a}_0,\mathbf{O}_T) + \Bigg[ \sum_{t=2}^{T-1} \log \frac{q(\mathbf{X}_{t-1}|\mathbf{X}_t,\mathbf{X}_0,\mathbf{O}_T)}{p_\theta(\mathbf{X}_{t-1}|\mathbf{X}_t)} \\
&\qquad + \log \frac{q(\mathbf{X}_{T-1}|\mathbf{a}_T,\mathbf{X}_0,\mathbf{O}_T)}{p_\theta(\mathbf{X}_{T-1}|\mathbf{X}_T)} + \log \frac{q(\mathbf{a}_T|\mathbf{X}_0,\mathbf{O}_T)}{q(\mathbf{X}_1|\mathbf{X}_0,\mathbf{O}_T)} \Bigg] + \log \frac{q(\mathbf{X}_1|\mathbf{X}_0,\mathbf{O}_T)}{p_\theta(\mathbf{X}_0|\mathbf{X}_1)} \Bigg) \Bigg] \\
&= \mathbb{E}_{q(\mathbf{X}_{0:T-1},\mathbf{a}_T|\mathbf{O}_T)}\Bigg[ \log \frac{q(\mathbf{a}_T|\mathbf{X}_0,\mathbf{O}_T)}{p_\theta(\mathbf{a}_T|\mathbf{O}_T)} + \log q(\mathbf{O}_0|\mathbf{a}_0,\mathbf{O}_T) + \Bigg[ \sum_{t=2}^{T-1} \log \frac{q(\mathbf{X}_{t-1}|\mathbf{X}_t,\mathbf{X}_0)}{p_\theta(\mathbf{X}_{t-1}|\mathbf{X}_t)} + \log \frac{q(\mathbf{X}_{T-1}|\mathbf{X}_T,\mathbf{X}_0)}{p_\theta(\mathbf{X}_{T-1}|\mathbf{X}_T)} \Bigg] \\
&\qquad - \log p_\theta(\mathbf{X}_0|\mathbf{X}_1) \Bigg] \\
&= \mathbb{E}_{q(\mathbf{X}_{0:T-1},\mathbf{a}_T|\mathbf{O}_T)}\Bigg[ \underbrace{\log \frac{q(\mathbf{a}_T|\mathbf{X}_0,\mathbf{O}_T)}{p_\theta(\mathbf{a}_T|\mathbf{O}_T)}}_{L_T} + \log q(\mathbf{O}_0|\mathbf{a}_0,\mathbf{O}_T) + \sum_{t=2}^{T} \underbrace{\log \frac{q(\mathbf{X}_{t-1}|\mathbf{X}_t,\mathbf{X}_0)}{p_\theta(\mathbf{X}_{t-1}|\mathbf{X}_t)}}_{L_{1:T-1}} \underbrace{-\log p_\theta(\mathbf{X}_0|\mathbf{X}_1)}_{L_0} \Bigg]
\end{aligned}
$$

Here $L_T$ is constant and hence can be ignored. We minimize $\{L_t\}_{t=0}^{T-1}$ by factorizing $p_\theta(\mathbf{X}_{t-1}|\mathbf{X}_t) = p_\theta(\mathbf{a}_{t-1}|\mathbf{X}_t)p_\theta(\mathbf{O}_{t-1}|\mathbf{X}_t), \ \forall \ t \in \{1,\dots,T\}$, and subsequently $p_\theta(\mathbf{a}_{t-1}|\mathbf{X}_t)$ is formulated as $\mathcal{N}(\mu_{t-1}(\mathbf{a}_t,\mathbf{O}_t), \sigma_{t-1}\mathbf{I})$ where, when using the noising process without drift (as in (7)),

$$
\begin{aligned}
\mu_{t-1}(\mathbf{a}_t,\mathbf{O}_t) &= \mathbf{a}_t - \sqrt{1-\overline{\alpha}_t}\epsilon_\theta(\mathbf{a}_t,\mathbf{O}_t,t) \\
\sigma_{t-1} &= \sqrt{1-\overline{\alpha}_{t-1}},
\end{aligned}
\tag{9}
$$

and when using the noising process with drift (as in (6)),

$$
\begin{aligned}
\mu_{t-1}(\mathbf{a}_t,\mathbf{O}_t) &= \frac{\sqrt{\overline{\alpha}_{t-1}}}{\sqrt{\overline{\alpha}_t}} \left( \mathbf{a}_t - \sqrt{1-\overline{\alpha}_t}\epsilon_\theta(\mathbf{a}_t,\mathbf{O}_t,t) \right) \\
\sigma_{t-1} &= \sqrt{1-\overline{\alpha}_{t-1}}.
\end{aligned}
\tag{10}
$$

Additionally, $p_\theta(\mathbf{O}_{t-1}|\mathbf{X}_t)$ is formulated as a non-differentiable *masking* or *zooming* process.

### A.3  Noising processes

#### A.3.1  Noising process with drift

Let $\mathbf{a}_0$ be a random variable over which we wish to define a generative distribution. We define inference distributions over latents $\{\mathbf{a}_1,\dots,\mathbf{a}_T\}$ as

$$
q(\mathbf{a}_t|\mathbf{a}_{t-1}) = \mathcal{N}(\mathbf{a}_t; \sqrt{\alpha_t}\mathbf{a}_{t-1}, \sqrt{1-\alpha_t}I),
\tag{11}
$$

where $\{(1-\alpha_i)\}_{i=1}^{t}$ is a fixed noise schedule.

We can unroll this recursion to obtain the distribution over $\mathbf{a}_t$ for any $t$ given the sample $\mathbf{a}_0$, as follows:

$$
\begin{aligned}
\mathbf{a}_t &= \sqrt{\alpha_t}\mathbf{a}_{t-1} + \sqrt{1-\alpha_t}\epsilon_{t-1} \\
&= \sqrt{\alpha_t\alpha_{t-1}}\mathbf{a}_{t-2} + \sqrt{1-\alpha_t\alpha_{t-1}}\overline{\epsilon}_{t-2} \\
&= \dots \\
&= \sqrt{\overline{\alpha}_t}\mathbf{a}_0 + \sqrt{1-\overline{\alpha}_t}\epsilon,
\end{aligned}
\tag{12}
$$

where $\epsilon \sim \mathcal{N}(\mathbf{0},I)$.

**Proof of recursion:**

$$\sqrt{\alpha_t}\mathbf{a}_{t-1} + \sqrt{1-\alpha_t}\epsilon_{t-1}$$
$$= \sqrt{\alpha_t\alpha_{t-1}}\mathbf{a}_{t-2} + \sqrt{\alpha_t(1-\alpha_{t-1})}\epsilon_{t-2} + \sqrt{1-\alpha_t}\epsilon_{t-1}$$

Here $\epsilon_{t-1}$ and $\epsilon_{t-2}$ are $\mathcal{N}(\mathbf{0},I)$ Gaussian distributions. Their weighted sum will result into a Gaussian distribution whose variances are added up. That is,

$$\sqrt{\alpha_t(1-\alpha_{t-1})}\epsilon_{t-2} + \sqrt{1-\alpha_t}\epsilon_{t-1}$$
$$= \sqrt{\alpha_t(1-\alpha_{t-1}) + 1 - \alpha_t}\bar{\epsilon}_{t-2}$$
$$= \sqrt{1-\alpha_t\alpha_{t-1}}\bar{\epsilon}_{t-2}$$

where $\bar{\epsilon}_{t-2} \sim \mathcal{N}(0,I)$. We observe here that as $t \to \infty$, $\mathbf{a}_t \sim \mathcal{N}(0,I)$.

### A.3.2 Noising process without drift

We fix the inference distribution over the latent variables $\{\mathbf{a}_1, \ldots, \mathbf{a}_T\}$ as a diffusion process without the scaling term. That is, we define

$$q(\mathbf{a}_t|\mathbf{a}_{t-1}) = \mathcal{N}(\mathbf{a}_t; \mathbf{a}_{t-1}, \sqrt{1-\alpha_t}I), \tag{13}$$

where $\{(1-\alpha_i)\}_{i=1}^{t}$ is a fixed noise schedule.

We can unroll this recursion to obtain the inference distribution over any $a_t$ directly given the sample $a_0$, as follows:

$$
\begin{aligned}
\mathbf{a}_t &= \mathbf{a}_{t-1} + \sqrt{1-\alpha_t}\epsilon_{t-1} \\
&= \mathbf{a}_{t-2} + \sqrt{2-(\alpha_{t-1}+\alpha_t)}\,\bar{\epsilon}_{t-2} \\
&= \ldots \\
&= \mathbf{a}_0 + \sqrt{t - \sum_{\tau=1}^{t}\alpha_\tau}\,\epsilon,
\end{aligned}
\tag{14}
$$

where $\epsilon \sim \mathcal{N}(\mathbf{0},I)$. We observe here that as $t \to \infty$, $\mathbf{a}_t \sim \text{Unif}(-\infty,\infty)$.

*Note:* If we analyze the noising process in (6) as $t_k \to T$, since $\overline{\alpha}(t_k) \to 0$ we have $\tilde{\mathbf{a}}_k^{(i)} \to \epsilon_k^{(i)}$. This means that if we train our denoising network $\epsilon_\theta$ to predict $\epsilon_k^{(i)}$, around $t_k \to T$ the model is trained to model an identity function (Salimans & Ho, 2022). This hampers training and eventual model performance. We point out that the noising process without drift does not suffer from this issue. Additionally, we also found it useful to directly tune $\overline{\alpha}$ (as opposed to tuning the underlying $\beta$-schedule) to prevent identity-training issues.

## B  Tasks and their Desiderata

**Sweeping-piles.** This task requires the model to output the parameters of a push primitive, that is the starting location and orientation of a pusher and ending location, to push piles of small objects into a target zone.

**Place-red-in-green.** This is a slightly more precision-requiring task where the robot is supposed to pick up a red 4x4x4 cm block using a suction cup and place it in a green bowl. The table is also laid with distractor blocks that can hinder model precision leading to imprecise picks. We also test our model for robustness against unseen distractor objects in this task by replacing blocks with daily-life objects such as pen, remote, stapler, etc.

**Kitting-part.** In this task, 5 parts of a skateboard truck assembly are laid out on a tabletop, and the robot is tasked to grasp the truck base on its slotted end and place correctly in a kit. To succeed in the task, our

Table 5: Simulation ($\star$) and real-robot ($\dagger$) tasks and their desiderata.

| Task | Ignore distractions | Precision | Position tolerance |
|---|---|---|---|
| **sweeping-piles**$^\star$ | ✗ | ✗ | n/a |
| **place-red-in-green**$^\star$ | ✓ | ✓ | 4.0 cm |
| **kitting-part**$^{\star\dagger}$ | ✓ | ✓ | 2.0 cm |
| **hang-cup**$^{\star\dagger}$ | ✗ | ✓ | 1.0 cm |
| **two-part-assembly**$^\star$ | ✗ | ✓ | 0.5 cm |
| **place-block-in-bowl**$^\dagger$ | | | |
| - big blocks | ✓ | ✗ | 4.5 cm |
| - small blocks | ✓ | ✓ | 2.5 cm |
| **screw-insert**$^\dagger$ | ✗ | ✓ | 1.0 cm |

hypothesis is that the robot needs to fixate on and observe the part in detail, while ignoring distractions that can hinder accuracy.

**Hang-cup.** This task requires the model to predict the pose of the gripper for picking and placing that should result in the cup hanging by its handle on the hook of a Y-shaped hanger. In the best possible scenario where the cup is oriented such that the handle's plane is perpendicular to the hanger's hook, the tolerance for error in position prediction is only 1 cm.

**Two-part-assembly.** This is a high-precision assembly task where the robot is tasked to pick up a skateboard truck hanger and insert it in the hole of the truck base. This task is a subroutine of assembling a full skateboard truck.

# C   Model Architecture and Training Details

We use a deep convolutional neural network with residual connections (ResNet-18 He et al. (2015)), without any pretraining, to process images, the output of which we then flatten and process using an MLP with two hidden layers ($\text{enc}_\phi$). We concatenate the encoder embeddings with query actions, which are then further processed by 4 fully-connected feedforward layers with skip connections ($\epsilon_\theta$). We use ReLU activations for all intermediate layers. We implemented all baselines with the same backbone architecture, tune learning rate in the range $[10^{-4}, 10^{-3}]$, and train using the Adam optimizer (Kingma & Ba, 2014) with a batch size of 100 demonstrations on a single Nvidia 2080 Ti GPU. We use rotation augmentation of the demonstrations.

# D   Hyperparameters

Table 6: Hyperparameters used for simulation experiments. In case of multiple entries, the one in **bold** worked best.

| Hyperparameter | Diffusion Policy | C3DM-Mask (ours) | C3DM (ours) |
|---|---|---|---|
| **Num noisy action samples** | 1, **10** | 1, **10** | **1**, 10 |
| **Noise schedule** | linear | linear | linear, linear |
| **Timestep encoding size** | 64 | 64 | 64 |
| **Downsample ratio** | N/A | N/A | 0.2 |
| **Learning rate** | 1e-4 | 1e-4 | 1e-4 |

Table 7: Hyperparameters used for real robot experiments. In case of multiple entries, the one in **bold** worked best.

| Hyperparameter | Diffusion Policy | C3DM-Mask (ours) | C3DM (ours) |
|---|---|---|---|
| **Num noisy action samples** | 1, 10, **50** | 1, 10, **50** | **1**, 10, 50 |
| **Noise schedule** | linear | linear | linear, $\mathbf{cos}^2$ |
| **Timestep encoding size** | 64 | 64 | 64 |
| **Downsample ratio** | N/A | N/A | 0.4 |
| **Learning rate** | 1e-4 | 1e-4 | 1e-4 |

Table 8: Hyperparameters used for sim-to-real robot experiments.

| Hyperparameter | Diffusion Policy | C3DM (ours) |
|---|---|---|
| **Num noisy action samples** | 1 | 1 |
| **Noise schedule** | linear | linear |
| **Timestep encoding size** | 256 | 256 |
| **Downsample ratio** | N/A | 0.28 |
| **Learning rate** | 1e-4 | 1e-4 |

## E   Unseen objects used in simulation experiments

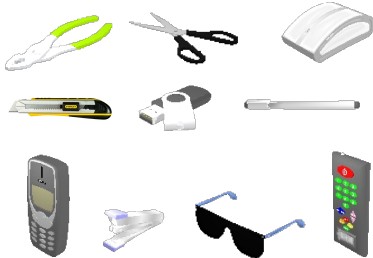

Figure 13: Table-top objects, unseen during training for any experiment, that we used to evaluate C3DM on ignoring novel distractor objects.

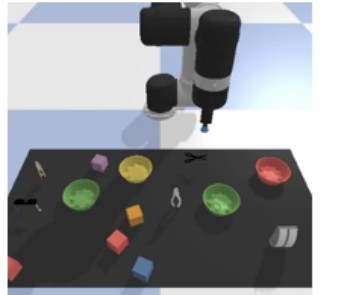
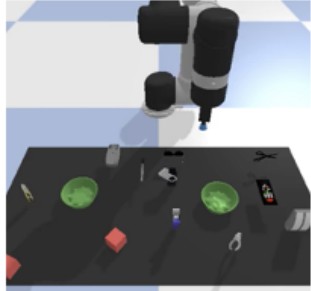

Figure 14: Table setup showing the *place-red-in-green* task with novel distractor objects.

## F   Details on motion planner used for tasks in simulation

For all tasks considered in Section 5.1, we implemented a motion planner that calculates a "hover" pose over the pick and place poses, and made the robot reach the hover pose both before and after attempting to

reach the corresponding predicted pose during action rollout. This hover pose was computed using a relative translation along the $+z$ axis on the predicted pick or place pose. To generate fine-grained motion, we did not obtain the collision model of the table-top, rather used linear interpolation to obtain the trajectories between target poses.

