# OpenReview forum: "C3DM: Constrained-Context Conditional Diffusion Models for Imitation Learning"
_TMLR — Accepted by TMLR_

### Review · Reviewer_nRSU · 2024-04-06

**Summary Of Contributions:**

This paper considers the problem of distractor objects during imitation learning. In particular, it notes that many modern methods are fragile in real-world deployment because of ever-changing objects in the environment that are irrelevant to the task at hand. Because extant models are powerful, they often learn spurious correlations. This paper considers a new method for learning representations that are robust to such spurious correlations, which they name Constrained Context Conditional Diffusion models. The main contribution of the C3DM pipeline is a so-called “fixation step”. This step acts somewhat as a data augmentation strategy and somewhat as an alignment step between image-space and the robot’s action space. Images are cropped around the central actuator in different ways. The cropped images are subsequently fed into a ResNet and then a type of diffusion policy tries to infer the correct actions from these augmented features. The method seems to work fairly well on a variety of settings, including real robot experiments and Sim2Real transfer. In particular, it seems to outperform Diffusion Policy in these settings.

**Audience:**

Yes

**Broader Impact Concerns:**

None.

**Claims And Evidence:**

Yes

**Requested Changes:**

- Please clarify the importance of depth cameras in obtaining the final results.
- Clarifying figures 6 and 7 would also go a long way towards improving the paper.
- Fix the notation in section 4.3.2 for fixation points.

**Strengths And Weaknesses:**

Strengths:

- The idea of inference through observations augmented with fixation points is an interesting one.
- The method seems to work well with its success rate often being substantially higher than Diffusion Policy.
- Real robot experiments are presented. Sim2Real is considered, and it seems that even in this setting, the robustness carries over.
- The motivation for solving this problem is clear.
- I like the formatting of Figure 5 with the tasks shown beneath their bar charts.

Weaknesses:

- It took me quite a long time to understand how fixation points are determined. I had to go to the algorithm box and read that several times and then go back to section 4.3.2 and read that several times. None of this was really spelled out clearly in the introduction or earlier parts of the paper, which spent a lot of energy talking about how to infer actions from observations. While this type of inference is interesting, and these graphical models are fine, I spent the entire time while reading these sections wanting to understand what in the world a fixation point is and where it comes from. I think section 4.3.2 does not make it particularly clear that a fixation point depends on the action. Although I can see now clearly that it is a function of the action, I think there’s something about the  img-T-real notation that is very off-putting. In particular, the text on both sides of the superscript made it difficult for me to understand what was going on. I would prefer this is changed in future drafts.

- It seems like ResNet is used for image encoding. While I realize this is a standard choice and perhaps beyond the scope of this paper, a lot of recent evidence suggests there are much better choices for the vision backbone. In fact, I would say ResNet is rather old-fashioned. See Masked Visual Pre-Training for Robotics for some more discussion and some possible models, but at the very least one would hope to see something like DINO as a potential vision backbone. Perhaps I’m missing something that makes this not possible, in which case I apologize.

- Figures 6 and 7 are extremely difficult for me to understand. Though I spent quite a long time thinking about what they’re showing and even asked a colleague if he understood what these figures were trying to demonstrate, I was ultimately unable to come to any solid conclusions. I can somewhat understand the density of arrows in figure 6, however that is about as far as I got.

- It seems that several of the experiments used a depth camera, but I wasn’t sure when the depth camera was being deployed and how important it was. The authors themselves seem to admit that height maps may be an important signal for many tasks. While this is not a major complication, I left the paper a little unsure how important depth cameras are to facilitating this work. If other people were to build on this work in the future, I think it is important that they understand if depth cameras are required to get solid results.

- As far as baselines, I think Diffusion Policy is fine. Why was Diffuser not included? I also am curious about more simplistic policies. For example, what if you collected five trajectories and next augment this data set by translating and zooming in on specific parts of the image? After that, train an MAE on the augmented data set. MAE, of course, works by masking certain parts of the image space, similar to your approach. Then you could use the MAE features to try and learn a diffusion policy directly. While I suspect this would still learn some spurious correlations, it would be good to verify that this sort of simplistic baseline does not work.

- Overall, I was left wondering how important is the alignment provided by fixation points between the image space and the observation space? Is that important, or is the diffusion pipeline for inferring actions from fixated observations more important? Put another way, can equation 4 still work out if we replace fixated obs with any other noisy observation. For example, directly adding Gaussian noise to the input image. I would like to see a more clear distinction between the work that was done in section 4.2 and the decisions that were made in section 4.3.2, as right now it is difficult for me to tell which part of this process is responsible for performance gains. While I do realize the paper has been written is such a way that these processes are inexorably linked, I do not think that this needs to be the case, and you should be able to ablate these components. Doing so would likely improve the paper significantly, because it could increase the number of baselines that are considered.

---

> ### Author Response · Authors · 2024-05-13
> **Response to Reviewer nRSU (1/4)**
>
> We thank the reviewer for their time spent in going through our paper and giving insightful comments. Please find below our response to their concerns.
>
> > I think section 4.3.2 does not make it particularly clear that a fixation point depends on the action. Although I can see now clearly that it is a function of the action, I think there’s something about the img-T-real notation that is very off-putting. In particular, the text on both sides of the superscript made it difficult for me to understand what was going on. I would prefer this is changed in future drafts.
>
> Thank you for pointing out the lack of clarity with regards to the selection of fixation point during training. The fixation process enables the model to focus on “action-relevant” parts of the scene, and hence is set to be a function of the target action during training, and the intermediate denoised action during testing. The former implements q(O_{t-1} | X_t, X_0) for training while the latter implements p(a_{t-1} | X_t) for testing. We have updated our paper to mention this explicitly in Section 4.3.2 for training as well as in Section 4.3.4 for testing.
>
> Additionally, img_T_real denotes a transformation that converts points in the camera frame to the image frame (as we mention in Section 4.3.2). This transform can also be written as $T_{img \leftarrow real}$, but we chose the former as it is quite standard to post-multiply a vector with a matrix in the space mentioned on the right of the transformation. If the reviewer  believes that this notation is confusing, we are happy to make the change throughout the paper.
>
> > It seems like ResNet is used for image encoding. While I realize this is a standard choice and perhaps beyond the scope of this paper, a lot of recent evidence suggests there are much better choices for the vision backbone. In fact, I would say ResNet is rather old-fashioned. See Masked Visual Pre-Training for Robotics for some more discussion and some possible models, but at the very least one would hope to see something like DINO as a potential vision backbone. Perhaps I’m missing something that makes this not possible, in which case I apologize.
>
> Thank you for bringing this up. ResNet was the model choice for a majority of our baseline methods that used images as observations (Diffusion Policy and IBC). During the architecture search for this paper, we tried ResNet-18 and ResNet-50 as visual encoders but did not find any difference in performance using either, and hence decided to stick with the smaller ResNet-18. This choice also made us more confident about our baseline performance as changing the core architecture usually requires hyperparameter tuning and since the baselines used ResNet-18 as well we were sure that they would perform as presented in their respective papers facilitating fair results. We want to point out that we prioritized keeping the base architecture for all the methods used in this paper exactly the same so as to ensure a fair comparison of the methodology without any effects due to architecture differences.
>
> Additionally, we are aware of the paper “Masked Visual Pre-Training for Robotics” which introduces a pre-training mechanism using a masked auto-encoder (MAE) for learning visual encodings, which are then frozen and utilized for visuomotor policy learning. As the reviewer suggested, DINO can also be utilized in a similar fashion to train a visual encoder in a self-supervised fashion on additional data, and then later fine-tuned/frozen for visuo-motor policy learning. We agree that these architectural changes with self-supervised learning could have advantages for improving generalization in visuomotor policies when deploying robots in environments with large variations. This is something we would certainly like to focus on in future work for scaling up this method for few-shot generalization to new tasks. However, we believe that such pre-training mechanisms that utilize additional data for added generalization are orthogonal to the objective of our current work and might introduce confounders to our evaluation. Hence we believe that our design choices for C3DM best showcase the differences in performance due to the different methodologies by keeping the datasets in-domain across all tasks, and neural network parameterizations same across all baselines and also matching in their architectures with their respective open-sourced models for the most fair comparison.

---

> > ### Comment · Reviewer_nRSU · 2024-05-14
> > **ResNets are not orthogonal**
> >
> > The use of ResNets is claimed to be orthogonal to the main advancements of this work. But I've seen a lot of work in this area where results completely vanish as soon as a better model such as DINO is used for the image backbone. I don't think its fair to call it orthogonal if using DINO completely equalizes all models.
> >
> > Such as it is, I'd feel quite nervous publishing this work with only a ResNET backbone, as I'd be afraid that all the shown results wouldn't hold up as soon as a stronger image backbone is used.
> >
> > Other than this one thing, I'm comfortable with this paper.

---

> > > ### Author Response · Authors · 2024-05-22
> > > **Response to reviewer (addressing concern regarding visual backbone)**
> > >
> > > Thank you for your response. To address your concern, we ran experiments using the DINOv2 backbone with pretraining (frozen and fine-tuning) on the place-red-in-green task in simulation. Below are the results for the models’ peak performance in 500 epochs of training.
> > >
> > > | **Method** | **ResNet-18 backbone** | **DINOv2 backbone (frozen)** | **DINOv2 backbone (fine-tuned)** |
> > > |--|:--:|:--:|:--:|
> > > | Diffusion Policy | 43 | 0 | 1 |
> > > | Diffusion Policy w/ zooming augmentation | 45 | 0 | 0 |
> > > | C3DM (ours) | 92 | 0 | 21 |
> > >
> > > The training curves for the Diffusion Policy and C3DM models can be found at this link: https://github.com/tmlr2024c3dm/tmlr2024c3dm/blob/main/Training%20losses%20with%20different%20visual%20backbones.pdf
> > >
> > > As can be seen in these results, **using a higher capacity visual backbone (DINOv2) with pre-training does not help these methods for visual manipulation,** even with the random zooming augmentation applied to the baseline. This is also in line with existing works that have shown that some recent approaches that leverage frozen visual representations [1] trained on large vision datasets are not useful for visual manipulation due to the large domain gap between pre-training datasets and the manipulation task [2]. From a model capacity point of view, we believe that the drop in success rate can be attributed to the model overfitting to the low amount of data that is available during training, rendering **ResNet-18 to be the better visual backbone for this setting.** We also want to point out that even with the DINOv2 backbone that we show is unsuitable for this data, our method still maintains a higher success rate over Diffusion Policy, making the ordering of models valid regardless of the visual backbone.
> > >
> > > Please let us know if these results address the only concern you had with the paper. We will be more than happy to answer any more questions you may have.
> > >
> > >
> > > References:
> > >
> > > [1] R3M: A Universal Visual Representation for Robot Manipulation, Nair et al., 2022
> > >
> > > [2] On Pre-Training for Visuo-Motor Control: Revisiting a Learning-from-Scratch Baseline, Hansen et al., 2023

---

> ### Author Response · Authors · 2024-05-13
> **Continuation of Response to Reviewer nRSU (2/4)**
>
> > Figures 6 and 7 are extremely difficult for me to understand. Though I spent quite a long time thinking about what they’re showing and even asked a colleague if he understood what these figures were trying to demonstrate, I was ultimately unable to come to any solid conclusions. I can somewhat understand the density of arrows in figure 6, however that is about as far as I got.
>
> Thank you for pointing out the lack of clarity in Figures 6 and 7. Figure 6 shows the fixation process on the input observation that C3DM facilitates in conjunction with the action denoising process. As we summarize in Algorithm 2, the fixation-while-denoising diffusion process (fDDP) computes the “fixation point” and the “fixated obs for next iter” at each denoising timestep. Figure 6 illustrates exactly those steps in fDDP where the fixation point (denoted by “x” in the figure) is obtained by denoising a noisy action using the predicted score function $\epsilon_{\theta}$, which is depicted in Figure 6 using a field of arrows. All the arrows in this field point toward a point that, after transforming from real to image space, is interpreted as a point in the image that the model should fixate on. Hence, at the next denoising timestep we present the model with a new observation that has been zoomed (for Figure 6) around the fixation point (“x”) in the previous timestep. Figure 6 lays out the score field, the induced fixation point, and the zooming mechanism used for fixation for the pick-and-place tasks. Throughout the fixation process we observe that the model is distracted at first with the entire table-top in view. As it zooms in, our model fixates closer and closer to the objects of interest showcasing a less distracted prediction owing to the removal of distractors and higher spatial resolution utilized during the denoising process.
>
> Figure 7, in contrast to Figure 6, shows the latent (noisy) actions (denoted by “^”) as they are iteratively denoised during fDDP. This figure also contrasts the two different mechanisms that we proposed for observation fixation in this paper, masking and zooming. We observe in this figure a qualitative example of how the latent actions are denoised into a predicted action in conjunction with the observation fixation process in fDDP.
>
> We have added more details to the captions of Figures 6 and 7 in the revised paper to make the figure more understandable to the reader. We also made the figures 10% larger for visual clarity.
>
> > It seems that several of the experiments used a depth camera, but I wasn’t sure when the depth camera was being deployed and how important it was. The authors themselves seem to admit that height maps may be an important signal for many tasks. While this is not a major complication, I left the paper a little unsure how important depth cameras are to facilitating this work.
>
> Thank you for pointing this out. We agree that additional discussion is needed to motivate the use of both RGB and depth modality.  To begin with, we note that our method is invariant to the input image modality (RGB or depth), as long as it allows for spatial context constraining. For all our experiments, except for sim-to-real transfer, we used RGB images as input to the model as a default setup. When attempting sim-to-real transfer, we switched to using depth maps (converted to height maps) to reduce the domain gap between sim and real observations, as echoed by a large body of sim-to-real research. We have added a statement in Section 5 of the revised paper to explicitly state this.

---

> ### Author Response · Authors · 2024-05-13
> **Continuation of Response to Reviewer nRSU (3/4)**
>
> > As far as baselines, I think Diffusion Policy is fine. Why was Diffuser not included? I also am curious about more simplistic policies. For example, what if you collected five trajectories and next augment this data set by translating and zooming in on specific parts of the image? After that, train an MAE on the augmented data set. MAE, of course, works by masking certain parts of the image space, similar to your approach. Then you could use the MAE features to try and learn a diffusion policy directly. While I suspect this would still learn some spurious correlations, it would be good to verify that this sort of simplistic baseline does not work.
>
> Thank you for your suggestions regarding baselines. Diffuser is a model-based goal-conditioned RL method that aims to solve manipulation problems using a diffusion-based planner in a low-dimensional state space. In contrast, our method focuses on offline learning from demonstrations in a purely supervised fashion, without goal conditioning, and using images as inputs. Due to these differences we believe a comparison with Diffuser would have been misleading. All our methods are supervised learning methods that learn to maximize the log-likelihood of training data (image inputs and low-dim output actions) in the learned conditional distribution over actions, and model the action space by either explicitly outputting the parameters of the action distribution (Conv. MLP), or implicitly as a distance (NGDF [1]) or score (Diffusion Policy [2], C3DM) field.
>
> While C3DM-Mask and C3DM do add the benefit of inherent data augmentation during training, they also provide the model with the ability to “fixate” on action-relevant parts of the scene during policy rollout. While we think that pre-training the visual encoder in an MAE fashion is an interesting idea, the objective to improve generalization using additional data is orthogonal to our work. However, to address your concern, we added additional results where we trained on an augmented dataset of zoomed observations and corresponding action coverage with Diffusion Policy and the explicit Conv. MLP baselines. This experiment appends baselines with all data augmentation that C3DM implicitly does as part of the fixation strategy when applied in conjunction with action denoising. As these results show, our method still outperforms baselines even when they were trained on additional data of observations and actions. We attribute this to the fact that our method actively makes effective use of the fixated observations in a structured manner during the iterative refinement process, unlike baselines that simply treat it as data augmentation and have no capability of ignoring distractions for solving the tasks at hand during policy rollout.
>
> Additional results (in **bold**) along with existing results for comparison:
>
> | Method / Task | place-red-in-green | kitting-part | hang-cup | two-part-assembly |
> |-|:-:|:-:|:-:|:-:|
> |Conv. MLP | 6 | 5 | 0 | 0 |
> | **Conv. MLP w/ zooming augmentation** | **0** | **6** | **0** | **0** |
> | DiffusionPolicy | 45 | 25 | 5 | 32 |
> | **DiffusionPolicy w/ zooming augmentation** | **45** | **30** | **7** | **35** |
> | C3DM (ours) | 92 | 90 | 50 | 56 |

---

> ### Author Response · Authors · 2024-05-13
> **Continuation of Response to Reviewer nRSU (4/4)**
>
> > Overall, I was left wondering how important is the alignment provided by fixation points between the image space and the observation space? Is that important, or is the diffusion pipeline for inferring actions from fixated observations more important? Put another way, can equation 4 still work out if we replace fixated obs with any other noisy observation. For example, directly adding Gaussian noise to the input image. I would like to see a more clear distinction between the work that was done in section 4.2 and the decisions that were made in section 4.3.2, as right now it is difficult for me to tell which part of this process is responsible for performance gains.
>
> Thank you for bringing up this point. The alignment between action and observation spaces is what makes the fixation process on observations possible, which we’d like to point out is just one specific realization of our proposed framework. On a higher level, we believe that the core contribution of C3DM is the conjunction of observation fixation with the action denoising process, where either can be independently chosen. However, there are constraints on choosing in the fixation process for observations, as we discuss below.
>
> To discuss the ablation you suggested, if we replace the denoising in equation (4) with Gaussian noise prediction and subtraction from $\mathbf{O}_t$, this would mean that the corresponding noising process infers the observation $\mathbf{O}_T$ as pure white noise. This does not sit well with the fact that we have the entire observation $\mathbf{O}_T$ available to us for generating actions (which is the only observation that Diffusion Policy uses) that we do not use, as well as all action denoising steps use a sub-optimal observation (due the added Gaussain noise) instead of using a rather useful context. To put another way, we designed the observation “denoising” processes as fixation processes that have one end of the process ($\mathbf{O}_T$) fixed to be the given input observation. Additionally, our ablation study with C3DM-Mask tries to address exactly the point you mentioned, that is it distinguishes the design choice for observation denoising (fixation) from action denoising. C3DM-Mask provides an alternative to the zooming strategy for fixation which highlights the importance of the proposed fixation strategy in our final method. Additionally, we also want to point out that Diffusion Policy is a special case of our method where $\mathbf{O}_t$ for all t=0 to T are the same as the input observation.
>
> Overall, we want to state that the fixation process on observations is an independent component of our method that we choose within the constraint of using the entire observation available to us at least once, and make sure that the designed fixation process provides the action denoiser with more useful observations as action denoising progresses. C3DM-Mask ablates the design choice of the fixation strategy by zooming and helps highlight it’s pros, and a comparison with Diffusion Policy helps highlight the usefulness of using a fixation process on all observations in conjunction with action denoising.
>
> References:
>
> [1] Neural Grasp Distance Fields for Robot Manipulation, Weng et al., 2022
>
> [2] Diffusion Policy: Visuomotor Policy Learning via Action Diffusion, Chi et al., 2023

---

### Review · Reviewer_Hcxb · 2024-04-28

**Summary Of Contributions:**

The paper presents an extension of diffusion models for imitation learning. To enhance its robustness, the authors propose to augment the model such that denoised actions are generated by conditioning on a restricted image (masked or zoomed in), which is itself generated based on the previous noisy action. By focusing on a restricted image, the method can be less impacted by spurious features.

**Audience:**

Yes

**Broader Impact Concerns:**

No applicable

**Claims And Evidence:**

Yes

**Requested Changes:**

To make the presentation clearer, I would suggest the authors to explain in more details the following points:
	- How is the integral in (2) computed?
	- How are the derivations in equation (14) obtained?

The paper contains a few minor issues:
	- Citation of authors should be either "author et al. (year)" or "(author et al., year)" depending if it's part of a sentence or not
	- Missing space before/after parentheses
	- Why does Figure 2 appear so early, while it is only referenced in the text on page 7?
	- Page 4: \theta = argmin

**Strengths And Weaknesses:**

STRENGTHS

The authors propose a novel extension of diffusion model.

This extension is well-justified under the assumption of observation-action alignment.

The experimental results both in simulation and on a real robot  system validate the approach.

WEAKNESSES

The presentation and writing could be improved (see below).

---

> ### Author Response · Authors · 2024-05-13
> **Response to Reviewer Hcxb**
>
> We thank the reviewer for their time spent in going through our paper and giving insightful comments. Please find below our response to their concerns.
>
> > How is the integral in (2) computed?
>
> Thank you for the question. We’d like to take the liberty of clarifying this in detail, and would be happy to take more questions if this is still unclear. As we note in Section 4.2, the aim of the model is to learn the parameters of the distribution $p_\theta(\mathbf{a}_0 | \mathbf{O}_T)$, where $\mathbf{a}_0$ is the observed action variable and $\mathbf{O}_T$ is a given observation of the table (an RGB/D image). The objective for learning the parameters of this distribution is to maximize the likelihood of data $(\mathbf{a}_0, \mathbf{O}_T)$ under the learned distribution. In eq. (2), we chose to introduce latent variables for both actions and observations in the distribution p(a_0 | O_T), similar to diffusion models or VAEs. These latent variables, as you pointed out, need to be marginalized out since they are not observed. However, the space of these latent variables is large and hence integrating them out exactly would be intractable, hence we compute the estimate of this integral by interpreting it as an expectation and compute the Monte Carlo estimate of this expectation as an average of samples. Furthermore, we introduce an auxiliary distribution q (our choice of this distribution is laid out in eq. (5)) which further helps with computing this estimate.
>
> > How are the derivations in equation (14) obtained?
>
> Thank you for the question. The derivation in equation (14) provides the proof that the noising process without drift (presented in the paper in equation (7)) leads to an action that would be sampled from a uniform random distribution when the timestep is large. The first line in the derivation is simply the noising process that obtains a_t from a_{t-1} by adding a Gaussian noise (equation (7)). Subsequently, if we use the same recursion on a_{t-1} we get a_{t-2} and an addition of two standard Gaussian variables whose variances can be added to obtain the second step in the derivation. We simply continue to unroll these same recurrence relation to obtain the final derivation in equation (14).
>
> > The paper contains a few minor issues: - Citation of authors should be either "author et al. (year)" or "(author et al., year)" depending if it's part of a sentence or not - Missing space before/after parentheses - Why does Figure 2 appear so early, while it is only referenced in the text on page 7? - Page 4: \theta = argmin
>
> Thank you for pointing out these issues. We went through the entire paper and fixed all citation issues following the format you recommended. Additionally, we added a reference to Figure 2 in the introduction to give the reader a look into the method details, which justifies the occurrence of Figure 2 early in the paper. We also fixed the equation as you pointed out in the Background section that needed us to replace min with argmin.

---

### Review · Reviewer_ACEe · 2024-04-29

**Summary Of Contributions:**

This paper presents C3DM, a diffusion-based model that predicts 2 6 DoF gripper positions for robotic pick-and-place tasks. During each diffusion iterative refinement step, the presented approach either zooms in or crops the model inputs (RGBD images) based on the current denoised outputs.

**Audience:**

Yes

**Broader Impact Concerns:**

No particular ethical or societal concerns

**Claims And Evidence:**

No

**Requested Changes:**

- While the paper is well motivated, the presented approach is underwhelming. Image or video generation tasks leverage the power of diffusion models as their output space is high dimensional. Here, the output space is only 12 dimensions, which counters the very motivation of using a diffusion model which sacrifices training and inference throughput. Can the authors comment on this? While one might argue that the iterative process enables opportunity for the proposed “fixation” approach to avoid “spurious correlation”, this claim is unjustified in the paper. Please see below for more details.
- Experiments
   - Can we have confidence intervals in table 2, 3 and 4?
   - In order to support the claim made in introduction, please include the following two baselines:
       1. behavior cloning without diffusion, same architecture.
       2. same as 1, but with the same amount of data augmentation (zooming and cropping) proposed in the approach.
   - I would be surprised if the proposed approach significantly outperforms baseline 2 above. But if it does and the author provide clear explanation with experiments/ablations, I would be happy to recommend acceptance.

**Strengths And Weaknesses:**

+ This paper is well written and clearly presented. I really like the simple and direct writing style.
+ The real-world manipulation experiment is a plus.

Please see below for weaknesses.

---

> ### Author Response · Authors · 2024-05-13
> **Response to Reviewer ACEe (1/2)**
>
> We thank the reviewer for their time spent in going through our paper and giving insightful comments. Please find below our response to their concerns.
>
> > While the paper is well motivated, the presented approach is underwhelming. Image or video generation tasks leverage the power of diffusion models as their output space is high dimensional. Here, the output space is only 12 dimensions, which counters the very motivation of using a diffusion model which sacrifices training and inference throughput. Can the authors comment on this?
>
> Thank you for pointing this out. While diffusion models have achieved major success in generating high-dimensional data such as images and videos, they have more recently proven to be a good model choice for robot learning as well [1-5]. Given the high cost of collecting expert demonstrations for imitation learning in robotics and the subsequent need to build sample-efficient algorithms, diffusion models, in contrast to standard behavior cloning architectures, have proven to be more sample-efficient and less reliant on hyperparameter tuning [1]. We agree with the reviewer that using a diffusion model sacrifices training and inference throughput, and which is what we claim that our method tries to trade off against increased sample efficiency and robustness to distractions. C3DM proposes a conditional action prediction framework that can learn to denoise actions while obtaining more task-relevant observations at each denoising iteration. The state-of-the-art methodology for doing this (Diffusion Policy [1]) utilizes a single observation and denoises a random action into one from the data distribution. Our method, on the other hand, learns a fixation process in conjunction with the action denoising process that allows the model to generate and use a more useful context for each action denoising step. As we showcase in our experiments, the fixation process in C3DM allows for improving robustness against distractions (in or out of distribution) as well as improving sample efficiency in real-world deployment (as shown in our real robot and sim-to-real experiments).
>
> > Can we have confidence intervals in table 2, 3 and 4?
>
> Thank you for pointing this out. Tables 3 and 4 showcase real robot experiments, and due to the limitations of performing real world experiments we were unable to add confidence intervals for those results, which is also in keeping with the related works [1-5] that do not show confidence intervals in real robot experiments. As you suggested, we have added confidence intervals for model performance in Table 2 using 3 random seeds. Below are the updated results showing mean +/- one standard deviation:
>
> | Method | 50% unseen | 100% unseen |
> | -------- | -------- | ------- |
> | Diffusion Policy | 41.7% +/- 5% | 33.3% +/- 10% |
> | C3DM (ours) | 85% +/- 4% | 90% +/- 4% |

---

> ### Author Response · Authors · 2024-05-13
> **Continuation of Response to Reviewer ACEe (2/2)**
>
> > In order to support the claim made in introduction, please include the following two baselines:
> > 1. behavior cloning without diffusion, same architecture.
> > 2. same as 1, but with the same amount of data augmentation (zooming and cropping) proposed in the approach.
> I would be surprised if the proposed approach significantly outperforms baseline 2 above. But if it does and the author provide clear explanation with experiments/ablations, I would be happy to recommend acceptance.
>
> We are happy to include additional results on pick-and-place tasks in simulation in support of our paper. To address (1) above, we note that in the paper we included a behavior cloning baseline without diffusion, trained using L2 regression which we called “Conv. MLP”. We want to point out that all the methods presented in our paper use the exact same neural architecture, including Conv. MLP. To address (2), we include results below using a version of Conv. MLP that was trained with the same amount of zooming and cropping augmentation as C3DM generates implicitly as part of the fixation strategy when applied in conjunction with the action denoising. As we notice, Conv. MLP baseline does not see any significant improvement in performance on any of the considered tasks even with the added data augmentation. Additionally, we also include results when training Diffusion Policy with this data augmentation. Here we find that Diffusion Policy does slightly benefit from the added zooming augmentation, however still does not surpass the performance of C3DM. To explain this, we note that these baseline methods, even though they have access to observations and actions at different zoom levels, do not possess any mechanism during inference to iteratively refine observations that can make active use of such augmentation. While baselines use a single observation, C3DM uses an observation fixation strategy (in fDDP) during iterative action denoising that effectively utilizes the additional data (implicitly generated) to output new observations and corresponding actions. The context-constraining process for generating observations ignores distractions by fixating on action-relevant parts of the input context and outputs a new context as input to the action denoising model at each refinement timestep. That is, our model is capable of making effective use of the augmentation during training in order to generate observations with different zoom levels and predict corresponding actions during testing, as summarized in the fDDP section of our method.
>
> Additional results (in **bold**) along with existing results for comparison:
>
> | Method / Task | place-red-in-green | kitting-part | hang-cup | two-part-assembly |
> |-|:-:|:-:|:-:|:-:|
> |Conv. MLP | 6 | 5 | 0 | 0 |
> | **Conv. MLP w/ zooming augmentation** | **0** | **6** | **0** | **0** |
> | DiffusionPolicy | 45 | 25 | 5 | 32 |
> | **DiffusionPolicy w/ zooming augmentation** | **45** | **30** | **7** | **35** |
> | C3DM (ours) | 92 | 90 | 50 | 56 |
>
> References:
>
> [1] Diffusion Policy: Visuomotor Policy Learning via Action Diffusion, Chi et al., 2023
>
> [2] Octo: An Open-Source Generalist Robot Policy, Ghosh et al., 2023
>
> [3] 3D Diffuser Actor: Policy Diffusion with 3D Scene Representations, Ke et al., 2024
>
> [4] Diffusion Meets DAgger: Supercharging Eye-in-hand Imitation Learning, Zhang et al., 2024
>
> [5] NoMaD: Goal Masked Diffusion Policies for Navigation and Exploration, Sridhar et al., 2023

---

> > ### Comment · Reviewer_ACEe · 2024-06-10
> >
> > Thanks for providing the additional experiments. The difference in performance is still much greater than I would have expected, which makes me think there might be a resolution problem in the original scale that are shared by the baseline, which is alleviated with the fact that the proposed method can zoom.

---

> > > ### Author Response · Authors · 2024-06-12
> > > **Response to reviewer**
> > >
> > > Thank you for your comment. As we showed in the additional results in our previous comment, C3DM performs better on all pick-place tasks even when the baselines methods (Conv. MLP and Diffusion Policy) are trained with the same amount of zooming and cropping augmentation as C3DM generates implicitly as part of the fixation strategy when applied in conjunction with the action denoising. To explain this, we noted that C3DM uses an observation fixation strategy (in fDDP) during iterative action denoising (action rollouts) that effectively utilizes the additional data (implicitly generated) to output new observations and corresponding actions, which a simple zooming data augmentation in the baselines cannot capture.
> > >
> > > > there might be a resolution problem in the original scale that are shared by the baseline, which is alleviated with the fact that the proposed method can zoom.
> > >
> > > We believe the reviewer means to say that they believe the baseline performances can be attributed to the low image resolution that the methods use in contrast to the higher resolution inputs that C3DM uses while zooming (please correct us if we misunderstood your comment). This concern was also raised by Reviewer A1im which we addressed by showing additional results using Diffusion Policy trained on multiple image resolutions, down from the image resolution used in the paper, upto the maximum possible resolution that matches with that of the cached high-res image used by C3DM for zooming in. We present those results below.
> > >
> > > We trained Diffusion Policy on the place-red-in-green task using different image resolutions, 160x160, 400x400, and 800x800, where the results on 160x160 inputs are those reported in the paper, and the results on 800x800 inputs are to highlight the performance of the baseline method when it has access to the same input resolution as C3DM.
> > >
> > > | Method | 160x160 input | 400x400 input | 800x800 input |
> > > |---|---|:---:|:---:|
> > > | Diffusion Policy | 43% | 31% | 19% |
> > > | C3DM | 92% (with fixation that uses cached 800x800 input) | - | - |
> > >
> > > The results above prove that using a higher resolution image can actually hurt the performance of the baseline during rollouts as it can exacerbate the learning of spurious correlations leading to more adverse effects due to distractors in the scene. C3DM makes effective use of the higher resolution context available to it by fixating on just the parts of the context that are relevant for the action, and hence performs much better in the presence of distractors.
> > >
> > > We hope this addresses the concerns of the reviewer that the benefits of our method are not singularly linked to the available high-res images or to the effective zooming augmentation in C3DM. We would be more than happy to answer any more questions they may have.

---

### Review · Reviewer_A1im · 2024-04-29

**Summary Of Contributions:**

* The authors present a new formulation for performing action diffusion while at the same time dealing with fixated, changing input observations (achieved through zooming in or masking)
* The authors evaluate their method on a wide range of simulated and real-world robotic pick-and-place tasks

**Audience:**

Yes

**Broader Impact Concerns:**

None.

**Claims And Evidence:**

Yes

**Requested Changes:**

* In Algorithm 2 the notation is different from Algorithm 1. In particular, the conditioning on $^{(i)}$ is missing
* Mention more clearly in Section 4.3.2 that the context is a function of the action
* A more thorough discussion regarding the comparison with diffusion policy in terms of offline trajectory generation before the movement vs. online trajectory generation during the movement. This is especially important since a wrist camera, in the latter case, can naturally provide more zoomed-in views of the scene and, therefore, more spatial resolution, which might be crucial to performing some of the tasks.
* Ideally, the authors provide additional results when running Diffusion Policy and their method in an online fashion.
* More details w.r.t. the employed Motion Planner should be provided
* The claims on computational efficiency should be backed up with additional results or toned down.
* The authors should mention more consistently during the paper that both, RGB and depth are considered as input modality. Moreover, the paper would benefit if the authors could comment on why they used the specific modality for the respective task.

**Strengths And Weaknesses:**

Strengths:

* The proposed idea of observation fixation during action diffusion is interesting and a very good contribution
* In the Appendix, the authors provide more detailed, thorough derivations for their approach.
* Thorough evaluation of the proposed method and comparison with multiple baselines

Weaknesses:

* While most parts of the paper are very well-written and easy to follow, I personally feel that the paper would benefit if the authors would mention more clearly in Section 4.3, and in particular in Section 4.3.2 that the context is a function of the action. While this ultimately gets clear in the algorithm boxes and is also somewhat mentioned in 4.3.2 already, I think an additional and more explicit statement would help. Moreover, the paper would be easier to understand if the authors would also make clear that their approach, therefore, only necessitates diffusion on the action level.
* In my opinion, the evaluations and, in particular, the comparison with diffusion policy is somewhat suboptimal. The output of the proposed approach in the paper are two 6D poses, which represent the picking and the placing pose. Contrarily, diffusion policy typically outputs more fine-grained, longer action trajectories. While the presented approach pre-computes the 2 waypoints before starting execution, diffusion policy is typically run online, and the action trajectories are constantly updated. This difference should, in my opinion, be mentioned more clearly. This aspect is crucial as the results suggest that the "zooming in" aspect of the work does have a significant effect (as shown in Fig. 5). From my point of view, part of the performance gains of C3DM vs Diffusion policy are clearly attributed to the increased spatial resolution which is available to C3DM (during the zooming process). This impression is underlined by the fact that the C3DM-masking baseline performs substantially worse than C3DM and only slightly better than diffusion policy. In this regard, I want to point out that if a wrist camera is added to the setup and if the action inference is also performed online (i.e., during trajectory execution), due to the movement of the arm and the wrist camera getting closer to the object of interest, C3DM-masking as well as Diffusion Policy could also benefit from a higher spatial resolution when being closer to the object that is to be manipulated or the placing location, which potentially could significantly close the performance difference between these methods and C3DM (zooming). While I still believe that the proposed method and formulation should result in better generalization, the previously mentioned points should be discussed in the paper. Ideally, the authors should provide additional ablation experiments, including wrist-mounted cameras and doing trajectory generation online, i.e., during policy execution.
* Still related to the previous point, regarding the training of the proposed method and the baselines, I was wondering whether only the initial views have been used as data points, i.e., do 5 demonstrations only result in 5 training data points, or did the authors also train using intermediate waypoints during trajectory execution? This should be clarified.
* The authors mention in the Problem Setup at the end of Section 3 that the policies can be deployed on a robot using "any off-the-shelf motion planner". While for most of the top-down pick and place tasks, I agree with the statement, for the more delicate tasks of "hang-cup" and "two-part-assembly," the choice of motion planner and, in particular, how the placing pose is approached might have a significant impact on the performance. Therefore, it is essential that the authors provide further information about which motion planner has been used and how it has been set up.
* In line with the previous comment, I want to remark that the work would greatly benefit from the authors' open-sourcing of their code.
* In Section 4.3.2, under the point "C3DM", the authors mention that "not using the entire high-res image keeps the model activation size short and inference time low, which is crucial for robotics." I personally found this statement problematic as the proposed C3DM approach does include the additional overhead of zooming in and having to re-render. Due to the fact that the vision input, therefore, changes on every iteration during denoising, also the vision-encoder has to be evaluated in every iteration. Since, in my personal experience, evaluating the vision-encoder often consumes most of the time, I would be curious about the inference times of C3DM compared to the other approaches for which the vision input is not changing and for which, therefore, the encoding of the visual input only has to be performed once. Moreover, the authors do not provide any ablations with respect to the resolution of the images, and as far as I understood, the activation sizes are the same for all of the approaches.
* The experiment section 5 is missing some details and lacks clarity. In particular, in the introductory paragraph of Section 5, the authors mention the number of demonstrations used for the real robot experiments and for the sim-to-real-experiments. However, they do not mention the number of demonstrations for the simulation experiments. I, therefore, recommend moving this information about the number of demonstrations in the respective sub-sections to have everything clear and in one place.
* I personally was a bit confused that, suddenly, for some of the experiments, the authors leveraged a depth camera. I personally think that the paper would benefit if the authors mentioned already during the method that the context can either be an RGB image or a depth map. Moreover, I would like to understand how the authors decided for which task to use the RGB images and for which ones to exploit the depth camera. Potentially, to improve clarity, the authors could add in brackets whether the respective experiment was conducted using RGB or depth input.

---

> ### Author Response · Authors · 2024-05-13
> **Response to Reviewer A1im (1/3)**
>
> We thank the reviewer for their time spent going through our paper and giving insightful comments. Please find below our response to their concerns.
>
> > In Algorithm 2 the notation is different from Algorithm 1. In particular, the conditioning on (𝑖) is missing.
>
> The superscript (i) in Algorithm 1 denotes the index of the sample within the training dataset. We’ve added a clarification for that in Section 4.3.1 in case that was unclear. Algorithm 2 lays out the procedure for action denoising during testing which is why we skipped the (i) superscript in that algorithm block. To make it clear that Algorithm 2 is for testing only, we have clearly specified that in the title of the algorithm in the revised paper. Please let us know if that does not address your concern.
>
> > I personally feel that the paper would benefit if the authors would mention more clearly in Section 4.3, and in particular in Section 4.3.2 that the context is a function of the action ... the paper would be easier to understand if the authors would also make clear that their approach, therefore, only necessitates diffusion on the action level.
>
> Thank you for pointing out the lack of clarity with regards to the selection of fixation point during training. The fixation process enables the model to focus on “action-relevant” parts of the scene, and hence is set to be a function of the target action during training, and the intermediate denoised action during testing. The former implements q(O_{t-1} | X_t, X_0) for training while the latter implements p_\theta( a_{t-1} | X_t) for testing. We have updated our paper to mention this explicitly in Section 4.3.2 for training as well as in Section 4.3.4 for testing. Additionally, we want to point out that this choice for the fixation point does stem from the experimental setup of modeling a conditional generative distribution over 12-dimensional action variables (as mentioned in the Background (Section 3) of the paper), and hence helps directly facilitate such generation.
>
> > A more thorough discussion regarding the comparison with diffusion policy in terms of offline trajectory generation before the movement vs. online trajectory generation during the movement. This is especially important since a wrist camera, in the latter case, can naturally provide more zoomed-in views of the scene and, therefore, more spatial resolution, which might be crucial to performing some of the tasks.
>
> Thank you for the insightful comment. We noted in our paper that our work resides within a well-studied keyframe-action problem setup for pick-and-place action prediction [1-3], which as you pointed out computes two waypoints before starting execution given the observation and does not do online trajectory generation. However, unlike cited works, our method does not assume a spatially-quantized action space but rather models a more complex continuous 6DoF action space, as noted in Section 1, closer to the baselines considered in our paper (e.g. Diffusion Policy) which are also apt comparisons given the contributions in generative modeling for action prediction in our work. The pick-and-place setup, however simple, is quite powerful and can be used for solving a variety of useful tasks (such as sweeping, kitting, assembly, etc.) as we have shown in the paper, even using Diffusion Policy which has previously been tested for online trajectory generation by its authors.
>
> As we point out in Section 4.3.2, C3DM benefits from the wide coverage of observations and actions during the training process, which our method implicitly generates during the training process. The zooming approach gives the model access to higher levels of detail during training as well as testing, an aspect which, as the reviewer pointed out, could further help online execution with Diffusion Policy. However, we’d still like to point out some key differences between the observation-conditioned Diffusion Policy and our framework. Diffusion Policy, as illustrated in Fig. 3, uses a single observation per generated denoising sequence while our framework allows for changing the observation at each action denoising step. Specifically, C3DM proposes a structured method implementing “fixation” embedded in the action denoising process which our model leverages for “improving” its observation as it iteratively refines the output action. We believe that online trajectory execution can benefit from adopting our framework, which however will pose additional challenges that we aim to overcome in future work. For instance, adapting our framework to online trajectory execution would mean that the fixation process cannot necessarily be a masking/zooming process since the action diffusion model can choose an arbitrary denoising Markov chain of fixation points during iterative refinement, and observation refinement would not necessarily start from a top-down view of the scene. This would require us to generate multi-viewpoint observations on the fly during training and testing.

---

> ### Author Response · Authors · 2024-05-13
> **Continuation of Response to Reviewer A1im (2/3)**
>
> > regarding the training of the proposed method and the baselines, I was wondering whether only the initial views have been used as data points, i.e., do 5 demonstrations only result in 5 training data points, or did the authors also train using intermediate waypoints during trajectory execution? This should be clarified.
>
> Thank you for pointing this out. N training demos means N pairs of image-action labels, where the images are top-down views of the table-top. We do not train using any intermediate waypoints of the trajectory output by the motion planner for training. We have edited the paper in Section 5.1.1 to clarify this.
>
> > The authors mention in the Problem Setup at the end of Section 3 that the policies can be deployed on a robot using "any off-the-shelf motion planner". While for most of the top-down pick and place tasks, I agree with the statement, for the more delicate tasks of "hang-cup" and "two-part-assembly," the choice of motion planner and, in particular, how the placing pose is approached might have a significant impact on the performance.
>
> For all our tasks, we implemented a motion planner that calculates a “hover” pose over the pick and place poses by using a relative translation along +z axis on the predicted pose. The trajectory in between poses was computed as a linear interpolation between poses. Since motion planning is not the focus of this study, we did not obtain the collision model of the table-top and use an off-the-shelf motion planner to plan the trajectory, but we stand by the claim that one could employ any off-the-shelf motion planner without any change to the proposed algorithm.
>
> > Since, in my personal experience, evaluating the vision-encoder often consumes most of the time, I would be curious about the inference times of C3DM compared to the other approaches for which the vision input is not changing and for which, therefore, the encoding of the visual input only has to be performed once.
>
> Thank you for pointing this out. Diffusion Policy would only require a single pass to encode the observation ($\mathbf{O_T}$) and can cache visual encodings for reuse across all action denoising timesteps. However, our method requires a pass through the vision encoder at each denoising step during action inference since we generate fixated observations ($\mathbf{O_0}, \dots, \mathbf{O_{T-1}}$) along with latent actions at each denoising timestep. We have highlighted this trade-off in the Limitations section of our revised paper (Section 6) to note that our method trades off precise action prediction and robustness against distractors for computation time during action inference.
>
> Additionally, we understand that our claim for "not using the entire high-res image keeps the model activation size short and inference time low, which is crucial for robotics" comes out to be ambiguous due to the additional visual encoding overhead introduced by our method. Hence, we have amended our statement in Section 4.3.2 to mention that keeping the image resolution low keeps the inference time as low as possible for this method given the additional computational overload introduced due to observation encoding at each denoising timestep.
>
> > The experiment section 5 is missing some details and lacks clarity. In particular, in the introductory paragraph of Section 5, the authors mention the number of demonstrations used for the real robot experiments and for the sim-to-real-experiments. However, they do not mention the number of demonstrations for the simulation experiments. I, therefore, recommend moving this information about the number of demonstrations in the respective sub-sections to have everything clear and in one place.
>
> Thank you for pointing this out. We had mentioned the number of demonstrations used for simulation and real experiments in their respective sections 5.1.1 and 5.2.1, and for the sim-to-real experiments presented a table with model performance with varying number of demonstrations in Section 5.2.2. We had highlighted the number of training demos used for real experiments in the intro para of Section 5, but have now included those for simulation experiments too for completeness and clarity in the revised paper.

---

> ### Author Response · Authors · 2024-05-13
> **Continuation of Response to Reviewer A1im (3/3)**
>
> > I personally was a bit confused that, suddenly, for some of the experiments, the authors leveraged a depth camera. I personally think that the paper would benefit if the authors mentioned already during the method that the context can either be an RGB image or a depth map. Moreover, I would like to understand how the authors decided for which task to use the RGB images and for which ones to exploit the depth camera.
>
> Thank you for pointing this out. We agree that additional discussion is needed to motivate the use of both RGB and depth modality.  To begin with, we note that our method is invariant to the input image modality (RGB or depth), as long as it allows for spatial context constraining. For all our experiments, except for sim-to-real transfer, we used RGB images as input to the model as a default setup. When attempting sim-to-real transfer, we switched to using depth maps (converted to height maps) to reduce the domain gap between sim and real observations, as echoed by a large body of sim-to-real research. We have added a statement in Section 5 of the revised paper to explicitly state this.
>
> References:
>
> [1] CLIPort: What and Where Pathways for Robotic Manipulation, Shridhar et al., 2021
>
> [2] Perceiver-Actor: A Multi-Task Transformer for Robotic Manipulation, Shridhar et al., 2022
>
> [3] Transporter networks: Rearranging the visual world for robotic manipulation, Zeng et al., 2021

---

> > ### Comment · Reviewer_A1im · 2024-05-14
> > **Response to Authors**
> >
> > Thanks for the answers and clarifications.
> >
> > Regarding the details of the motion planner, I would like to see this information included in the paper's Appendix, since in case You do not plan to open-source the code, such information is crucial for reproducing the results.
> >
> > My only major concern that remains for the paper is (as already pointed out in my initial review), the gap in performance between the proposed "zooming" C3DM and the masked C3DM version, and, in particular, the fact that the performance improvements of C3DM-masked w.r.t. the DiffusionPolicy baseline are considerably less significant (as shown in Fig. 5). Therefore, I am still concerned that having access to the zoomed-in views (using "a cached high-resolution image of the same scene" (page 7)) might be the crucial ingredient to achieve higher success rates and to outperform the baselines. Put differently, it might be that large parts of the reported performance differences arise from the fact that the different methods have access to different observations, i.e., the proposed C3DM method has access to a cached high-resolution image, which is not available for the other methods. While the authors comment on this fact at the beginning of page 12 ("While C3DM-Mask was able to ignore distractions well, it lacked precision when evaluated with smaller blocks."), I feel it should be given greater importance in the Limitations and Conclusion section. In this regard, I still believe that the paper would greatly benefit if the authors would compare their proposed method with running DiffusionPolicy online such that the policy naturally has access to more close-up views of the object that is to be manipulated (when approaching the object). Alternatively, for a more fair comparison between the methods, they could also ablate the performance of their method if the zooming-in is done based on only the initial image that is also available to the other methods (i.e., removing the access to the "high-resolution image" for zooming in).
> >
> > I still believe that the presented paper is interesting and has its contributions. However, I feel that the comparison w.r.t. the baselines could be significantly improved, especially w.r.t. the fact that in the current comparison, the different methods have access to different observations.

---

> > > ### Author Response · Authors · 2024-05-22
> > > **Response to reviewer (addressing input resolution disparity concern)**
> > >
> > > We thank the reviewer for their response.
> > >
> > > We have added details regarding the motion planner in Appendix F of the revised paper.
> > >
> > > **Addressing reviewer’s concern regarding the disparity in available input resolution to C3DM and Diffusion Policy:**
> > >
> > > For context, the cached high-res image has a resolution of 800x800 which C3DM uses to obtain fixated “zoomed-in” observations around action-relevant parts of the scene. For all results in the paper, this high-res image is downsampled 5x to obtain the observation O_T for both C3DM and Diffusion Policy, i.e. an image of size 160x160 as input to all methods. C3DM can then query for “fixated observations” during the denoising process that are also 160x160 images (as shown in Figure 12 in Appendix A.1) obtained by “zooming in” and leveraging the high-res cached image.
> > >
> > > To address the reviewer’s concern, we trained Diffusion Policy on the place-red-in-green pick/place task using different image resolutions, 160x160, 400x400, and 800x800, where the results on 160x160 inputs are those reported in the paper, and the results on 800x800 inputs are to highlight the performance of the baseline method when it has access to the same input resolution as C3DM.
> > >
> > > | Method | 160x160 input | 400x400 input | 800x800 input |
> > > |---|---|:---:|:---:|
> > > | Diffusion Policy | 43% | 31% | 19% |
> > > | C3DM | 92% (with fixation that uses cached 800x800 input) | - | - |
> > >
> > > The results above prove that using a higher resolution image can actually hurt the performance of the baseline during rollouts as it can exacerbate the learning of spurious correlations leading to more adverse effects due to distractors in the scene. C3DM can make effective use of the higher resolution context available to it by fixating on just the parts of the context that are relevant for the action, and hence performs much better in the presence of distractors.
> > >
> > > We hope this addresses the concerns of the reviewer that the benefits of our method are not singularly linked to the high-res images. We would be more than happy to answer any more questions they may have.

---

> > > > ### Comment · Reviewer_A1im · 2024-05-27
> > > > **Additional Experiments**
> > > >
> > > > Thanks for the additional results that do address my previous concern.

---

### Decision · Action_Editor_zncn · 2024-06-22

**Recommendation:** Accept as is

**Comment:**

The paper considers the problem that arises when policies trained via imitation learning become sensitive to spurious correlations that result from changes in objects that are not relevant to the tasks. To address this, the paper proposes Constrained Context Conditional Diffusion Models (C3DM) the primary component of which is a fixation step that serves as both an alignment and augmentation approach using either a zoomed in or masked version of the input. The paper demonstrates the effectiveness of C3DM through experiments on both simulated and real-world domains.

Considering both the initial submission as well as the discussion between the authors and reviewers, there is general agreement that the paper not only considers an important problem that arrises when employing imitation learning, but that the means by which the proposed diffusion model-based method mitigates the effects of spurious correlations via its "fixation step" is both novel and interesting. Notably, the reviewers particularly appreciate the fact that the paper demonstrates the effectiveness of C3DM not only in simulation, but on real robots as well. While there were initially some questions regarding the significance of the contributions of C3DM over contemporary methods, the subsequent discussion between the authors and reviewers largely resolved these concerns.

**Audience:**

The paper is of interest to many in the TMLR community.

**Claims And Evidence:**

Following the author-reviewer discussion, the significance of the paper's contributions with regards to its proposal of a novel approach to performing action diffusion in the face of changing inputs is clear and sufficiently supported.

During the review process, several reviewers raised concerns about inadequate justification for some of the core claims made in the initial submission. Among them, these involved questions about the sufficiency of the experimental comparisons to baseline methods (e.g., Diffusion Policy), the validity of claims regarding the generalizability to different motion planners, and the extent to which performance gains can be attributed to conditioning on restricted images (e.g., by zooming in). The authors adequately addressed these concerns during the discussion phase.